

# The braincase of *Mesosuchus browni* (Reptilia, Archosauromorpha) with information on the inner ear and description of a pneumatic sinus

Gabriela Sobral[1] and Johannes Müller[2]

[1] Staatliches Museum für Naturkunde Stuttgart, Stuttgart, Germany
[2] Museum für Naturkunde Berlin, Leibniz-Institut für Evolutions- und Biodiversitätsforschung, Berlin, Germany

## ABSTRACT

Rhynchosauria is a group of archosauromorph reptiles abundant in terrestrial ecosystems of the Middle Triassic. *Mesosuchus* is one of the earliest and basalmost rhynchosaurs, playing an important role not only for the understanding of the evolution of the group as a whole, but also of archosauromorphs in general. The braincase of *Mesosuchus* has been previously described, albeit not in detail, and the middle and inner ears were missing. Here, we provide new information based on micro-computed tomography scanning of the best-preserved specimen of *Mesosuchus*, SAM-PK-6536. Contrary to what has been stated previously, the braincase of *Mesosuchus* is dorso-ventrally tall. The trigeminal foramen lies in a deep recess on the prootic whose flat ventral rim could indicate the articulation surface to the laterosphenoid, although no such element was found. The middle ear of *Mesosuchus* shows a small and deeply recessed fenestra ovalis, with the right stapes preserved in situ. It has a rather stout, imperforated and posteriorly directed shaft with a small footplate. These features suggest that the ear of *Mesosuchus* was well-suited for the detection of low-frequency sounds. The semicircular canals are slender and elongate and the floccular fossa is well-developed. This is indicative of a refined mechanism for gaze stabilization, which is usually related to non-sprawling postures. The most striking feature of the *Mesosuchus* braincase is, however, the presence of a pneumatic sinus in the basal tubera. The sinus is identified as originating from the pharyngotympanic system, implying ossified Eustachian tubes. Braincase pneumatization has not yet been a recognized feature of stem-archosaurs, but the potential presence of pneumatic foramina in an array of taxa, recognized here as such for the first time, suggests braincase sinuses could be present in many other archosauromorphs.

## INTRODUCTION

Rhynchosaurs were the first herbivorous archosauromorph reptiles (Fig. 1). Derived taxa show unique adaptations to herbivory such as bony beaks formed by the premaxilla and

Corresponding author
Gabriela Sobral,
gabriela.sobral@smns-bw.de

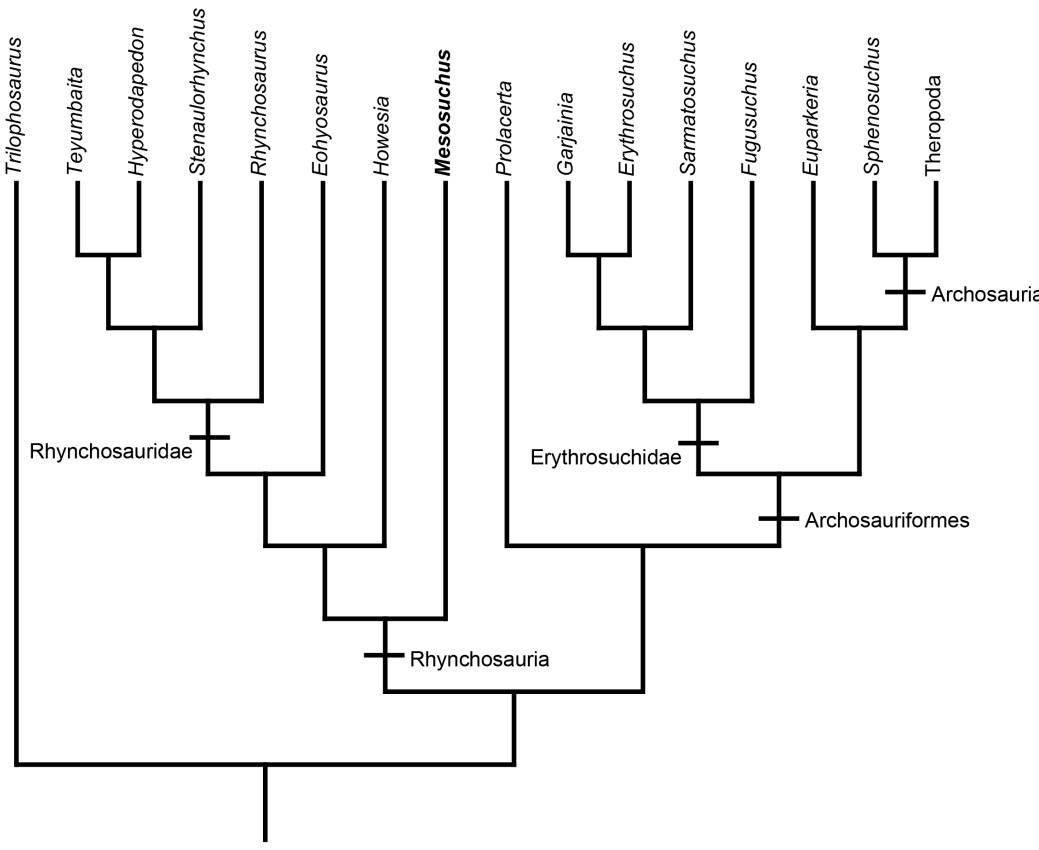

**Figure 1 Rhynchosaur phylogeny.** Phylogenetic relationships of relevant archosauromorph taxa, including *Mesosuchus* and other rhynchosaurs. Tree based on *Ezcurra (2016)* and *Ezcurra, Montefeltro & Butler (2016)*.

tooth plates with complex medio-lateral replacement suitable for processing hard plant material (*Benton, 1984*). Rhynchosaurs originated in the Early Triassic, in the aftermath of the end-Permian mass extinction (*Ezcurra, Montefeltro & Butler, 2016*), and became important faunistic elements of Middle and Late Triassic ecosystems (*Butler et al., 2015*). Most of the rhynchosaur diversity suddenly disappeared at around the Carnian-Norian boundary, playing an important role in pointing an extinction event at about this time (*Brusatte et al., 2010*).

*Mesosuchus* is one of the basalmost, and one of the earliest, rhynchosaurs known (Fig. 1). The recovered fossil material includes a nearly complete skeleton with an anatomy intermediate between other archosauromorphs and rhynchosaurids, making it an important taxon to understand the early evolution of the clade. *Mesosuchus* has also been important for the assessment of the interrelationships of stem-archosaurs in general. *Dilkes (1998)* recovered for the first time prolacertiforms (or protorosaurids) as paraphyletic, and Rhynchosauria was placed as the sister-group to *Prolacerta* + Archosauriformes. Subsequent studies, however, were not always congruent with this suggested position. For instance, *Modesto & Reisz (2002)*, *Müller (2004)*, and *Chen et al. (2014)* recovered *Prolacerta* as a more basal archosauromorph taxon than rhynchosaurs, while *Hill (2005)* found rhynchosaurs even outside of Sauria.

Until the re-evaluation of *Dilkes (1998)*, the anatomy of *Mesosuchus* was poorly understood. In his study, an extensive description of the braincase of *Mesosuchus* was provided, but some details pertaining to the middle and inner ears, as well as vascular and nervous elements, were either missing or presented as uncertain.

Here, we present a reassessment of the braincase anatomy of *Mesosuchus*, clarifying and correcting some more obscure points, as well as adding new data on the middle and inner ears based on high resolution micro-computed tomography (µCT) scans. We also discuss the palaeobiological implications derived from this new information, contextualizing rhynchosaurs in a broader archosauromorph scenario.

## MATERIALS AND METHODS

The *Mesosuchus browni* braincase used here is housed at the Iziko South African Museum under the catalog number SAM-PK-6536. The braincase was scanned in the Museum für Naturkunde Berlin using a Phoenix|x-ray Nanotom (GE Sensing and Inspection Technologies GmbH, Wunstorf, Germany). X-ray slices were reconstructed in the software datos|x-reconstruction version 1.5.0.22 (Phoenix|x-ray; GE Sensing and Inspection Technologies GmbH, Wunstorf, Germany) and the resulting data were segmented and analyzed in VG Studio Max 2.1 (Volume Graphics, Heidelberg, Germany). Scans comprised 1,440 projections, and were made with a tungsten target using a Cu filter of 0.1 mm thickness in modus 0, averaging 3, skip 2, and exposure time of 750 ms. The material was scanned with exposure time of 1,000 ms, voltage of 80 kV, current of 450 µA, and voxel size of 27.08 µm.

In addition, we scanned the skull of the rhynchosaur *Eohyosaurus wolvaardti* for comparative purposes. The material is also housed at the Iziko South African Museum, under the catalog number SAM-PK-K10159. The scanning parameters used for this material are the same as described above, except for the following: exposure time of 1,000 ms, 85 kV, 400 µA, and a voxel size of 22.79 µm.

Both CT scans are stored in the public digital collection of the Museum für Naturkunde Berlin and they can be accessed through the µCT lab (mikroctlabor@mfn-berlin.de) or, alternatively, through the authors. Additionally, the 3D model of the braincase of Mesosuchus can be found online under the address sketchfab.com/models/d332dbe1e6f8452ea8314d64989ddda7.

## DESCRIPTION

The braincase of *Mesosuchus* is displaced postero-ventrally in relation to the remaining of the skull, so that the brain cavity is visible in dorsal view, but concealed by the retroarticular process of the lower jaw in lateral view (Fig. 2). Except for the absence of a laterosphenoid (whose presence has never been confirmed for this taxon) and a crack on the left paroccipital process, the braincase is intact. Preservation and preparation, however, seem to have eroded, or at least weakened some parts of the braincase, as evidenced by the µCT scans. The braincase is dorso-ventrally tall, that is, the basipterygoid processes, basal tubera and occipital condyle do not lie in the same transverse plane *contra* the depiction of *Evans (1986*; fig. 9 therein*)* and the coding of *Ezcurra (2016)*—see below.

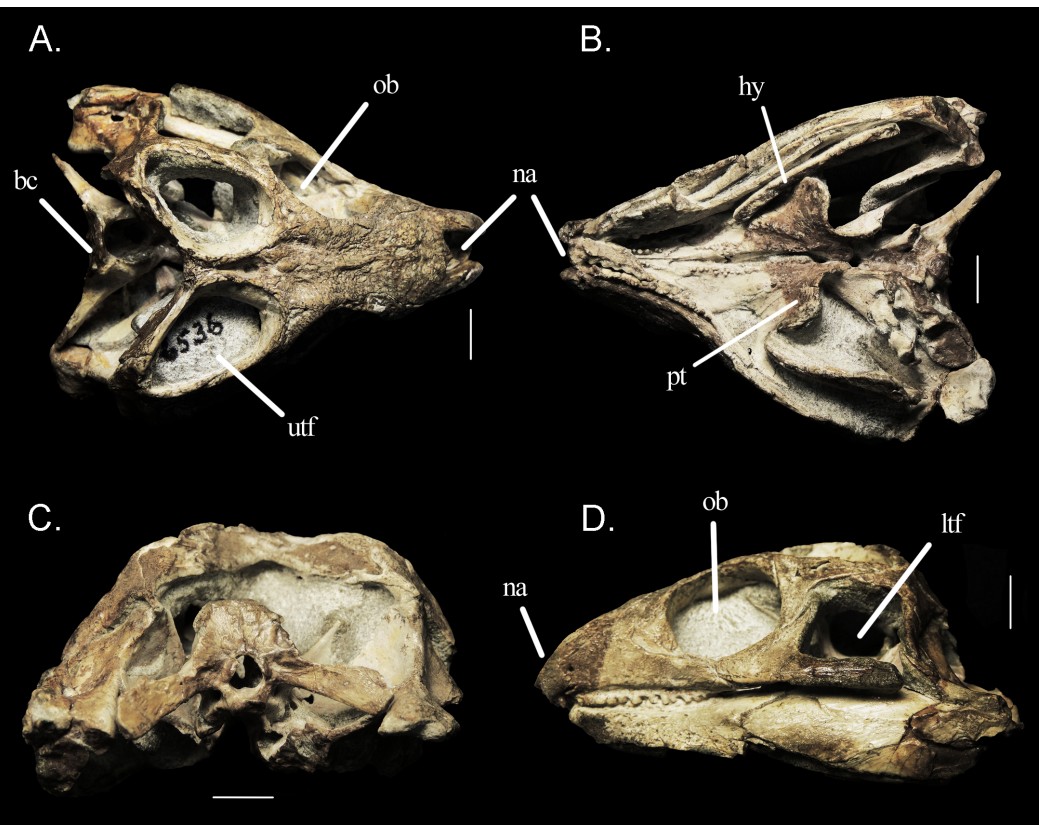

**Figure 2  Specimen SAM-PK-6536.** Skull of *Mesosuchus browni* in (A) dorsal, (B) ventral, (C) posterior, and (D) lateral views. Abbreviations: bc, braincase; hy, hyoid element; ltf, lower temporal fenestra; na, nare; ob, orbit; pt, pterygoid; utf, upper temporal fenestra. Scale bars equal one cm.

The **basioccipital** contributes to the ventral rim of the foramen magnum and to the floor of the foramen of cranial nerve (CN) XII. The occipital condyle faces posteriorly in comparison to the braincase floor. Contrary to *Dilkes (1998)*, we do not describe it as oval, but as sub-triangular in shape, with the apex pointing ventrally (Figs. 3A–D). Just anterior to the condyle, in ventral vew, the basioccipital shows a low, anteriorly concave rim connecting the bases of the left and right basal tubera, to which the basioccipital contributes to the posterior portion (Fig. 3). Acid preparation, however, seems to have eroded this contribution, probably exaggerating the height of the ridge. The ridge itself is similar to that of *Euparkeria* (*Sobral et al., 2016*). The ridge of *Mesosuchus* also delimits the posterior extension of the shallow median pharyngeal recess. The suture between basioccipital and parabasisphenoid runs around the postero-medial border of the bases of the tubera, but it is difficult to follow it entirely because the bones are dorso-ventrally thin. The suture follows a short way antero-medially while describing a smooth arch, which is posteriorly concave. On the right side, it shows no contribution to the opening foramen of the pharyngotympanic sinus (Eustachian tube—see below), but on the left side it seems to contribute to part of its posterior border. On the dorsal surface of the basioccipital, a low median ridge separates the right and left surfaces.

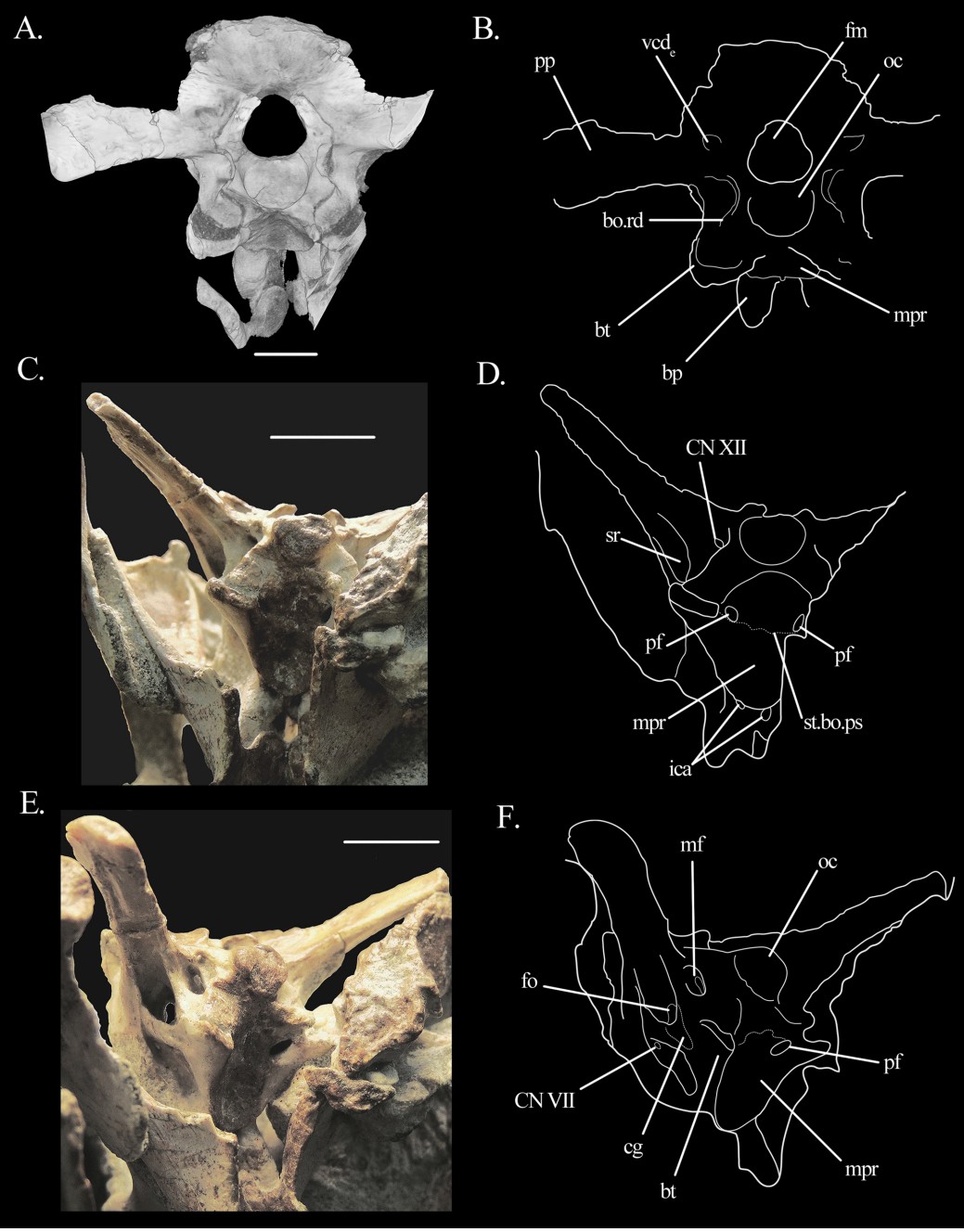

**Figure 3 Braincase of *Mesosuchus browni*.** (A) CT scan image in posterior view with (B) corresponding line drawing, (C) photograph in ventral view with (D) corresponding line drawing, and (E) photograph in latero-ventral view with (F) corresponding line drawing. Abbreviations: bo, basioccipital; bp, basiptergygoid process; bt, basal tuber; cg, closed gap; fm, foramen magnum; fo, fenestra ovalis; ica, internal carotid artery; mf, metotic foramen; mpr, median pharyngeal recess; oc, occipital condyle; pf, pneumatic foramen; pp, paroccipital process; ps, parabasisphenoid; rd, ridge; sr, stapedial recess; st, suture; vcd_e, external foramen of the vena capitis dorsalis. Roman numerals refer to corresponding cranial (CN). Scale bars equal five mm in (A) and one cm in (C) and (E).

The suture between para- and basisphenoid could only be identified in some parts of the braincase, but the element will be treated as a single unit, the **parabasisphenoid**. The cultriform process is long and possesses a sulcus running along the entire extension of its dorsal surface. The basipterygoid processes are prominent and directed slightly anteriorly. We could not identify the path of the maxillo-mandibular branch of the trigeminal nerve (CN V) on the antero-dorsal surface of the basipterygoid processes in this specimen, but the branch does not seem to have run through a canal as described by *Dilkes (1998)*. The "triangular gap" described by *Dilkes (1998)* seems to be used as a synonym for the "semilunar depression" of *Evans (1986)*, but its description seems to topologically correspond to the lateral depression (=anterior tympanic recess) of *Euparkeria* (*Sobral et al., 2016*). The semilunar depression of *Mesosuchus* will be dealt with in the discussion, but no lateral depression was identified in SAM-PK-6536. The parabasisphenoid contributes to the ventral border of the fenestra ovalis.

The entrance foramina for the cerebral branch of the internal carotid arteries are located on the ventral surface of the parabasisphenoid (Figs. 3C and 3D), posterior and medial to the bases of the processes. The canals run antero-dorsally and open on the floor of the hypophyseal fossa. They are not confluent. The fossa is relatively shallow because the clinoid processes are very low—lower than the dorsal margin of the cultriform process. The middle parts of the dorsum sellae are missing in SAM-PK-6536, but the remaining lateral portions are narrow and connect to the ventral rim of the anterior inferior process of the prootic (Fig. 4). A subtle ridge runs antero-posteriorly in the midline of the hypophyseal fossa, sub-dividing it into two (Fig. 4C). Neither a foramen nor a pathway for CN VI was found. The posterior half of the ventral surface of the parabasisphenoid is concave, forming the anterior part of the shallow median pharyngeal recess. The anterior limit of the recess is made by a shallow, posteriorly concave ridge, which contributes to the posterior borders of the foramina of the internal carotid artery (Figs. 3C and 3D). The outline formed by this ridge flares posteriorly and forms the (anterior) parabasisphenoid contribution of the basal tubera. *Dilkes (1998)* identifies this structure as "crista ventrolateralis." The anterior surfaces of the parabasisphenoid contributions to the basal tubera are somewhat flattened and they are excavated by the ventral border of the fenestra ovalis dorsally.

The **prootic** conforms with that of other archosauromophs. Posteriorly, it seems to contribute to a small antero-proximal portion of the paroccipital process, although the extent of this contribution cannot be quantified (Figs. 5A–5C). It forms the anterior border of the fenestra ovalis and, anteriorly, all but the antero-dorsal border of the foramen of CN V. The foramen is deeply recessed, especially in its posterior region (Figs. 4A and 4B), indicating the position of the gasserian ganglion. Interestingly, the flat, rectangular surface of the entire ventral rim of the foramen may be indicative of an articular surface, although no laterosphenoid was found. On the lateral surface of the prootic, there are three sets of crests (Figs. 4A, 4B, 5A and 5B). The first is the crista prootica (or otosphenoidalis). It runs antero-ventrally from the base of the paroccipital process, bending sharply ventrally at the antero-dorsal portion of the facial nerve (CN VII). By doing so, it divides the prootic into anterior and posterior surfaces, where the foramina

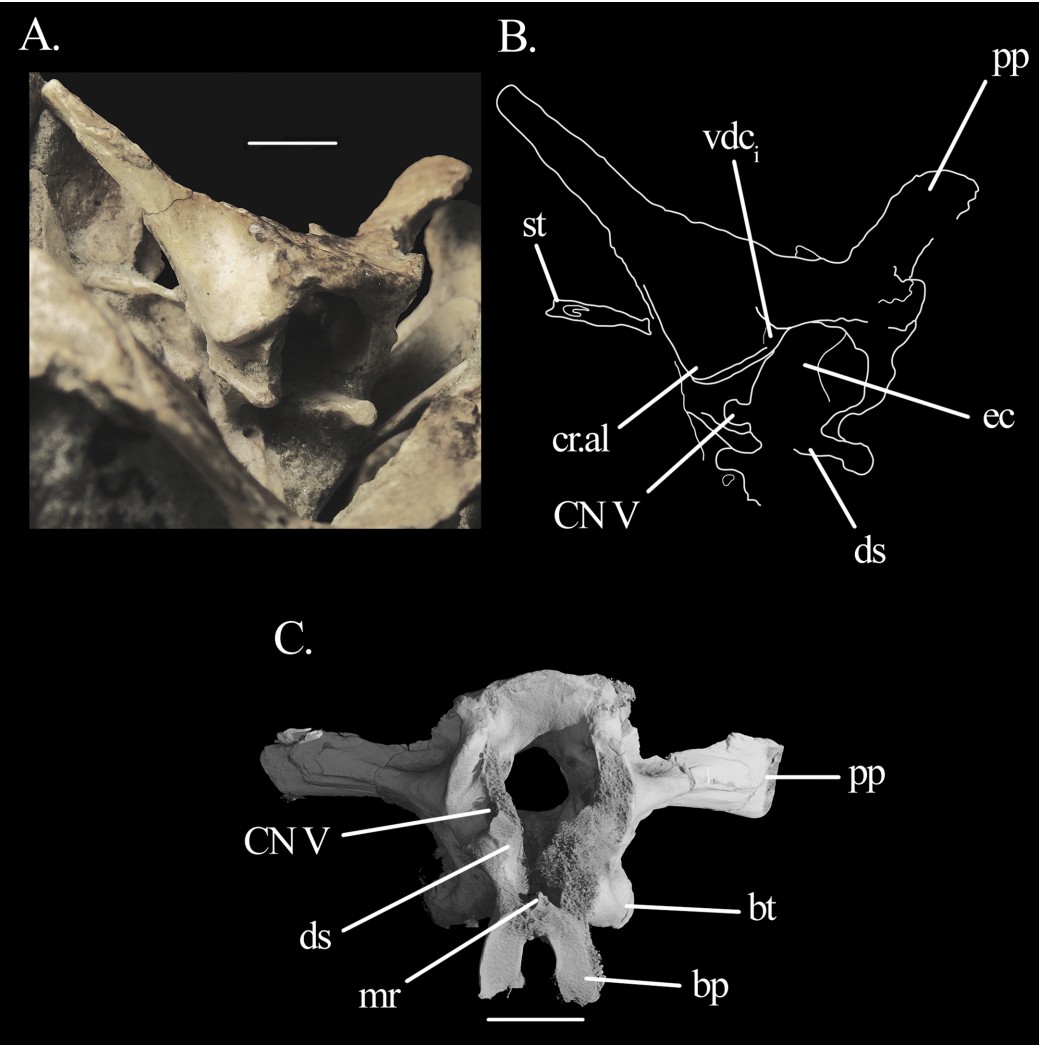

**Figure 4 Braincase of *Mesosuchus browni*.** (A) Photograph in antero-lateral view with (B) corresponding line drawing, and (C) CT scan image in anterior view. Abbreviations: bp, basipterygoid process; bt, basal tuber; cr.al, crista alaris; ds, dorsum sellae; ec, endocranial cavity; mr, median ridge; pp, paroccipital process; st, stapes; vcd$_i$, internal foramen of the vena capitis dorsalis. Roman numeral refers to corresponding cranial nerve (CN). Scale bar equals one cm in (A) and seven mm in (C).

for CN V and CN VII are found, respectively. The crista prootica continues ventrally onto the posterior border of the basipterygoid process. The second crest has no particular name. It is sharp and originates dorsal to the foramen of CN VII, running ventrally between it and the fenestra ovalis. It is posteriorly concave. The dorsal part of the crista prootica runs ventral to a third, rounded crest sometimes termed crista alaris. It runs from a point anterior to the ventral bending of the crista prootica sharply anterodorsally to the anterior margin of the prootic. It terminates dorsal to the foramen of CN V. The lateral surface of the prootic dorsal to the crista alaris is smoothly depressed.

On the antero-dorsal region of the prootic, close to the contact with the supraoccipital, there is an elevated structure that may represent the articulation facet of the parietal.

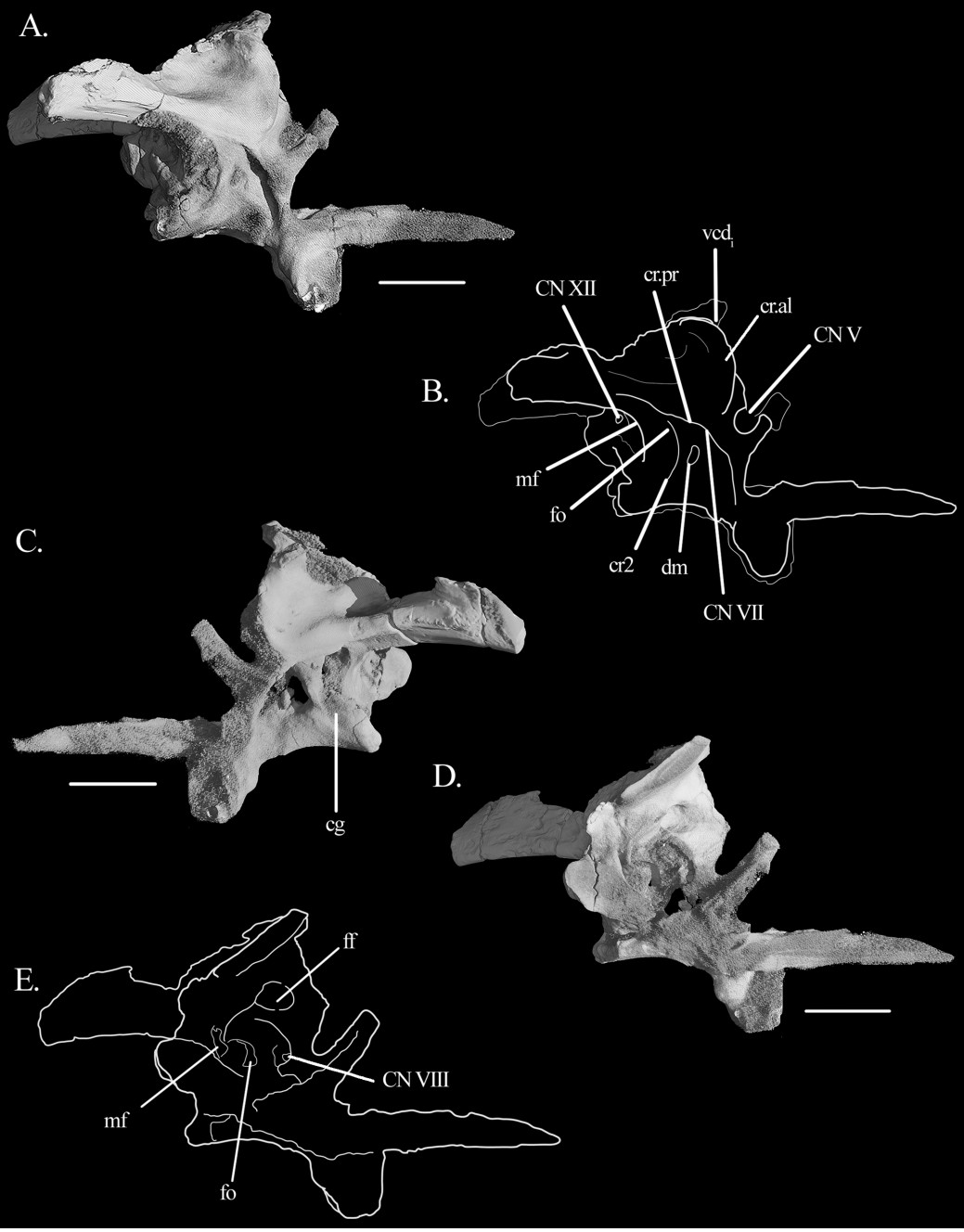

**Figure 5 Braincase of *Mesosuchus browni*.** Three-dimensional reconstruction of the braincase of *Mesosuchus browni* in (A) right lateral view with (B) respective line drawing, (C) left lateral view, and (D) left medial view with (E) corresponding line drawing. Abbreviations: cr2, "crest 2;" cr.al, crista alaris; cg, closed gap; cr.pr, crista prootica; dm, damage; ff, floccular fossa; fo, fenestra ovalis; mf, metotic foramen; vcd$_i$, internal foramen of the vena cpitis dorsalis. Roman numerals refer to corresponding cranial nerves (CN). Scale bars equal seven mm.

The area between this structure dorsally and the anteriormost limit of the third crest ventrally is depressed, indicating the internal foramen of the vena capitis dorsalis (Figs. 4A, 4B, 5A and 5B). The external foramen is found on the opisthotic. The prootic likely formed

the floor of this depression, with the supraoccipital forming the roof (see below). The foramen of CN VII is located posterior to the crista prootica, in a rather dorsal position, so that it is only visible in a more ventral view (Figs. 3E, 3F, 6A and 6B). The lateral surface of the prootic is damaged ventral to the foramen on both sides, and the bone is thinned in these areas. On the left side the damage is extensive enough to be confluent to the foramen (Figs. 5A, 5B, 6A and 6B). A shallow sulcus runs postero-dorsally from the foramen of CN VII along the ventral surface of the crista prootica and represents the hyomandibular branch of the facial nerve. Another shallow sulcus runs ventrally from the foramen of CN VII, indicating the proximal path of the palatine branch. The prootic forms the anterior border of the fenestra ovalis. On the left medial wall of the prootic, it is possible to see the much enlarged floccular fossa. The postero-ventral border of the fossa projects prominently medially and delimits the anterior region of the vestibule (Figs. 5D and 5E).

The **opisthotic** forms the posterior region of the lateral braincase wall (Fig. 5). No suture lines with the prootic or supraoccipital could be found, only with the exoccipital (see below). The opisthotic forms most of the paroccipital process. The processes are very long and antero-posteriorly flattened distally, with the anterior surface facing antero-ventrally. The ventral rim is straight, but the dorsal flares distally at the tip of the processes. They are strongly directed posteriorly and dorsally, at approximate angles of 45°. They lie on the same level as the foramen magnum. The proximal antero-ventral surface is extensively excavated by the recessus stapedialis (Figs. 3C–3F). The ventral ramus separates the fenestra ovalis from the metotic foramen and runs straight ventrally. Its distal part contacts the parabasisphenoid anteriorly and the basioccipital posteriorly. Contrary to what has been stated by *Evans (1986)*, it has no clubbed, triangular tip.

In posterior view, there are two small fossae on the opisthotic, dorsal to the paroccipital process and medially to it, close to the dorsal border of the foramen magnum (Figs. 3A and 3B). They likely represent the external foramina of the vena capitis dorsalis (see Discussion). On the medial surface, the opisthotic possesses a ventral expansion that represents the postero-dorsal wall of the vestibule. Ventral to it, there is the perilymphatic notch (Fig. 6E).

The **exoccipital** forms the posterior wall of the metotic foramen, most of the foramen of CN XII and the lateral border of the foramen magnum. In posterior view, the suture between exoccipital and opisthotic runs ventro-laterally to medio-dorsally, from the dorsal border of the metotic foramen to the dorsal border of the foramen magnum. The contact with the basioccipital is more difficult to detect. What seems to be a crack on the left side may indicate the position of the suture. It runs also ventro-laterally to medio-dorsally, following the floor of the foramen for CN XII. There are two openings for this nerve on the left side (Figs. 6A and 6B). They are difficult to detect with the naked eye, and are more easily distinguishable in the CT scans. They lie in a shallow, slit-like recess, also identified by *Dilkes (1998)*. The articulation facets for the proatlas are present as prominent projections.

The **supraoccipital** forms the braincase roof and contributes for a small part to the dorsal rim of the foramen magnum. The suture with exoccipital and opisthotic was not found, but could have been located in the area of the open spaces (see below). In lateral

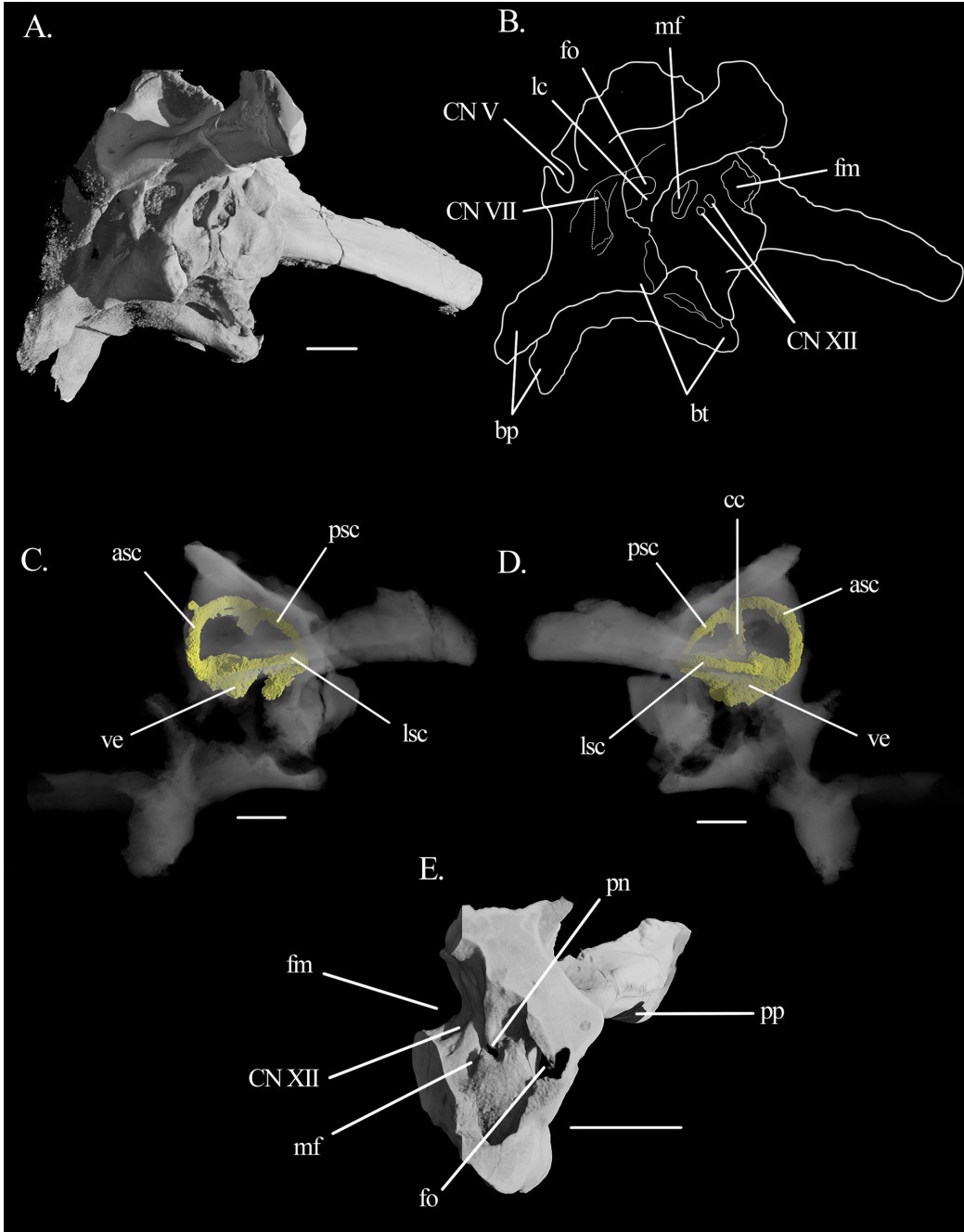

**Figure 6 Middle and inner ears of *Mesosuchus browni*.** (A) CT image of the middle ear of *Mesosuchus browni* in postero-ventral view with (B) corresponding line drawing; (C) inner ear in left lateral, and (D) right lateral views; (E) detail of the left medial side of the inner ear in antero-medial view. Abbreviations: asc, anterior semicircular canal; bp, basipterygoid processes; bt, basal tubera; cc, common crus; fo, fenestra ovalis; fm, foramen magnum; lc, lagenar crest; lsc, lateral semicircular canal; mf, metotic foramen; pn, perilymphatic notch; pp, paroccipital process; psc, posterior semicircular canal; ve, vestibule. Roman numerals refer to corresponding cranial nerves (CN). Scale bars equal 3.5 mm in (A), (C), and (D) and six mm in (E).           

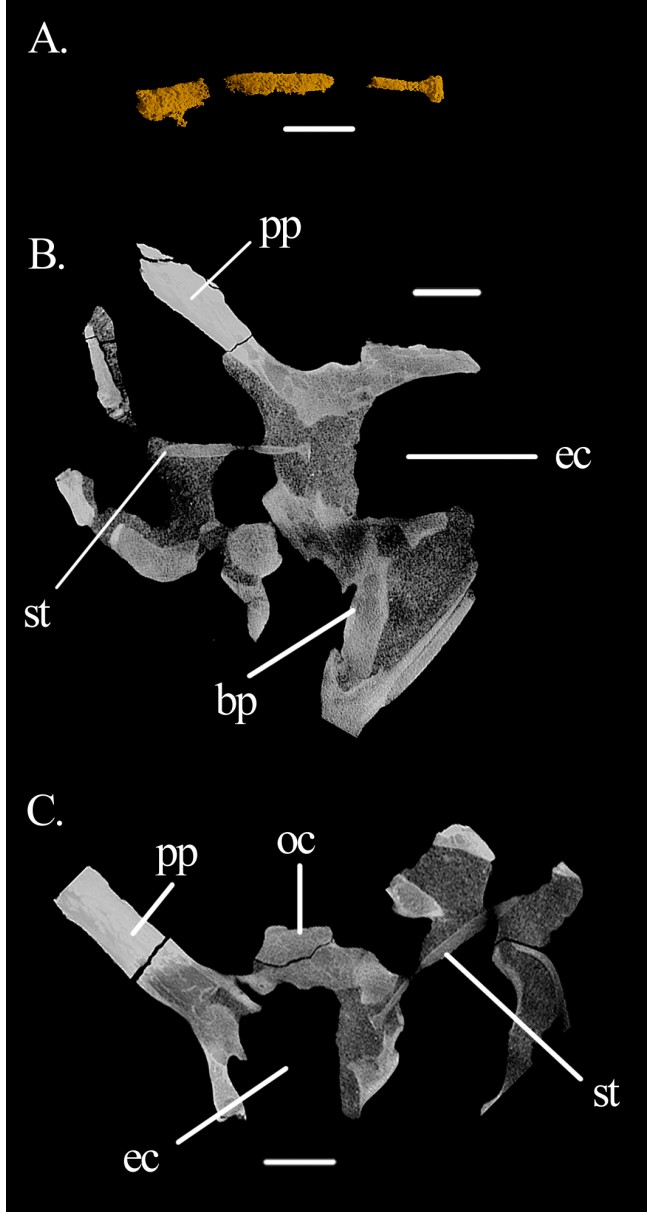

**Figure 7 Right stapes of *Mesosuchus bronwi*.** (A) 3D model of the segmented element; (B) coronal, and (C) transversal sections of the braincase with the stapes in articulation with the fenestra ovalis in anterior and ventral views, respectively (both views are slightly lateral to show the entire length of the stapedial shaft in the slice). Abbreviations: bp, basipterygoid process; ec, endocranial cavity; oc, occipital condyle; pp, paroccipital process; st, stapes. Scale bars equal 3.5 mm in (A), 4.5 mm in (B), and five mm in (C).

view, at the anteroventral tip of the supraoccipital there is a smoothly depressed area. This area represents the internal foramen of the vena capitis dorsalis, with a contribution from the prootic.

The **middle ear** has very distinctive features. The right stapes is preserved in situ (Figs. 4A, 4B, 7 and 8). The shaft is a long and rather stout rod that tapers proximally, forming a well-marked neck separating it from the footplate (Fig. 7A). The stapes is

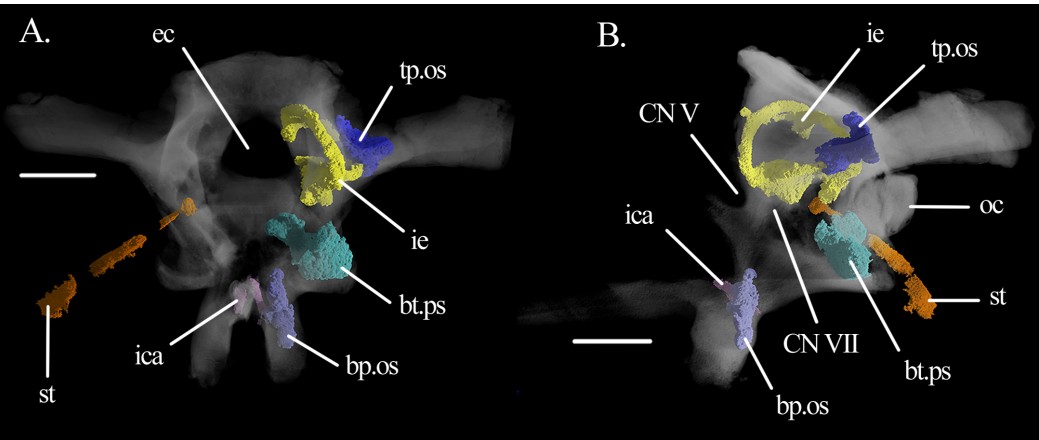

**Figure 8  3D model of open spaces.** Three-dimensional reconstruction of the open spaces of the specimen SAM-PK-6536 of *Mesosuchus browni* in (A) anterior and (B) left lateral views. Abbreviations: bp. os, open space in the basipterygoid process; bt.ps, basal tuber pneumatic sinus; ec, endocranial cavity; ica, internal carotid artery; ie, inner ear; oc, occipital condyle; st, stapes; tp.os, tympanic open space. Roman numerals refer to corresponding cranial nerves (CN). Scale bars equals 5.5 mm.

directed postero-ventrally (Fig. 7C). In dorsal view, it seems to follow the paroccpital process closely (Fig. 7C). The footplate is in articulation with the fenestra ovalis, filling it antero-posteriorly but not dorsoventrally (Figs. 7B and 7C). On the anterior surface, the stapes seems to possess a well-developed stapedial foramen on the shaft, but this could not be found in the CT scans, likely representing a preservational artifact only. The fenestra ovalis is dorsoventrally tall and anteroposteriorly short. On the left side, the lagenar crest can be seen on the mid-hight of its posterior border (Figs. 6A and 6B). The fenestra ovalis is quite small, and lies deep within a recess with sharp and broad rims. The recess excavates more than half of the antero-ventral surface of the paroccipital process and extends ventrally to the anterior (parabasisphenoid) surface of the basal tubera. As the ventral ramus of the opisthotic is slightly inclined postero-dorsally to antero-ventrally, the fenestra ovalis is located somewhat dorsal to the metotic foramen. The metotic foramen is big and its ventral part is wider than its dorsal region.

In the **inner ear** of *Mesosuchus*, the semicircular canals are elongate and slender compared to the vestibule (Figs. 6C and 6D). The anterior canal leaves the anterior ampulla antero-laterally and runs dorsally and postero-medially around the floccular fossa to the common crus. Only the dorsal part of the common crus lies within the supraoccipital, opisthotic, and prootic. However, reconstruction of the whole crus was possible because the right side is covered with matrix (Fig. 6D). The anterior semicircular canal is slightly longer than the posterior one. The lateral semicircular canal is the shortest of the three, leaving the anterior ampulla at its postero-lateral portion. It enters the utricular recess at about the same point as the posterior canal, which leaves the recess postero-laterally and then runs dorsally and antero-medially to the common crus. The medial wall of the braincase is mostly unossified, especially ventrally, but the anterior and postero-dorsal walls of the vestibule are present. The posterior branch of CN VIII

pierces the former (Figs. 5D and 5E). A prominent, smooth lagenar crest can be seen on the posterior border of the left fenestra ovalis (Figs. 6A and 6B). It separates the vestibule from the cochlear region at about the dorsal third of the height of the fenestra. The dorsal surface of the basioccipital in the cochlear region is somewhat concave, but shows no excavation for the cochlear recess (Fig. 7B), indicating the cochlea was short. On the medial surface, the ventral ramus of the left opisthotic bears a perilymphatic notch (Fig. 6E).

There is at least one **pneumatic sinus** in the braincase of *Mesosuchus* (Figs. 8 and 9), which is an unexpected feature for non-archosaur archosauromorphs. The braincase of *Mesosuchus* is trabeculate, which makes it difficult to assess the real extension of the sinus. The pneumatic sinus is located in the basal tubera and occupies most of them (Figs. 8 and 9A). It opens externally through a pair of foramina found close to the lateral borders of the ventral surface of the parabasisphenoid, just posterior to its contribution to the basal tubera. It likely originates from the pharygotympanic system (Eustachian system—see Discussion). The sinus has a puzzling communication with the brain cavity, as its dorsal portion is open medial, and slightly anterior to the metotic foramen, posterior to the inner ear proper (Fig. 8B). The medial and most of the posterior walls of the otic capsule in *Mesosuchus* were not ossified, but still a (soft tissue) separation between the cochlear region and the recessus scalae tympani would have existed. If this communication was present in life, there would likely have had a mechanic correlation with the inner ear. However, as such an anatomy has never, to our knowledge, been reported, and because we regard SAM-PK-6536 a mature individual (as discussed below), we think it is more likely that the area was composed of thin bone in life, and that the fossilization processes damaged the region.

Two other systems of open braincase spaces will be described here, although we are currently unsure if these correspond to real pneumatic sinuses (see Discussion). Part of these open spaces have smooth walls and are continuous with trabeculated regions. They may represent marrow spaces, but we decided to describe them here due to their partial resemblance with true pneumatic spaces, also in topology. The first of these open spaces is located within the basipterygoid processes (Figs. 9C and 9D). It seems to invade the base of the cultriform process and part of the parabasisphenoid body posterior to the processes, and anterior to the sinuses of the basal tubera without contacting them (Figs. 9E and 9F). The foramina connecting the space to a pneumatic system could not be confidently identified. The second open space lies within the opisthotic and potentially also in parts of the supraoccipital (Fig. 9B). It extends medio-laterally halfway into the paroccipital processes, enfolding the posterior part of the lateral semicircular canals dorsally, ventrally, and laterally. Its external foramina could also not be clearly identified.

Besides these braincase spaces, *Mesosuchus* possesses a medial pharyngeal recess on the ventral surface of the braincase floor (Fig. 3). The recess is delimited both anteriorly and posteriorly by low, concave ridges on the parabasisphenoid and basioccipital, respectively. We could not find evidence for the presence of an anterior tympanic recess *sensu Sobral et al. (2016).*

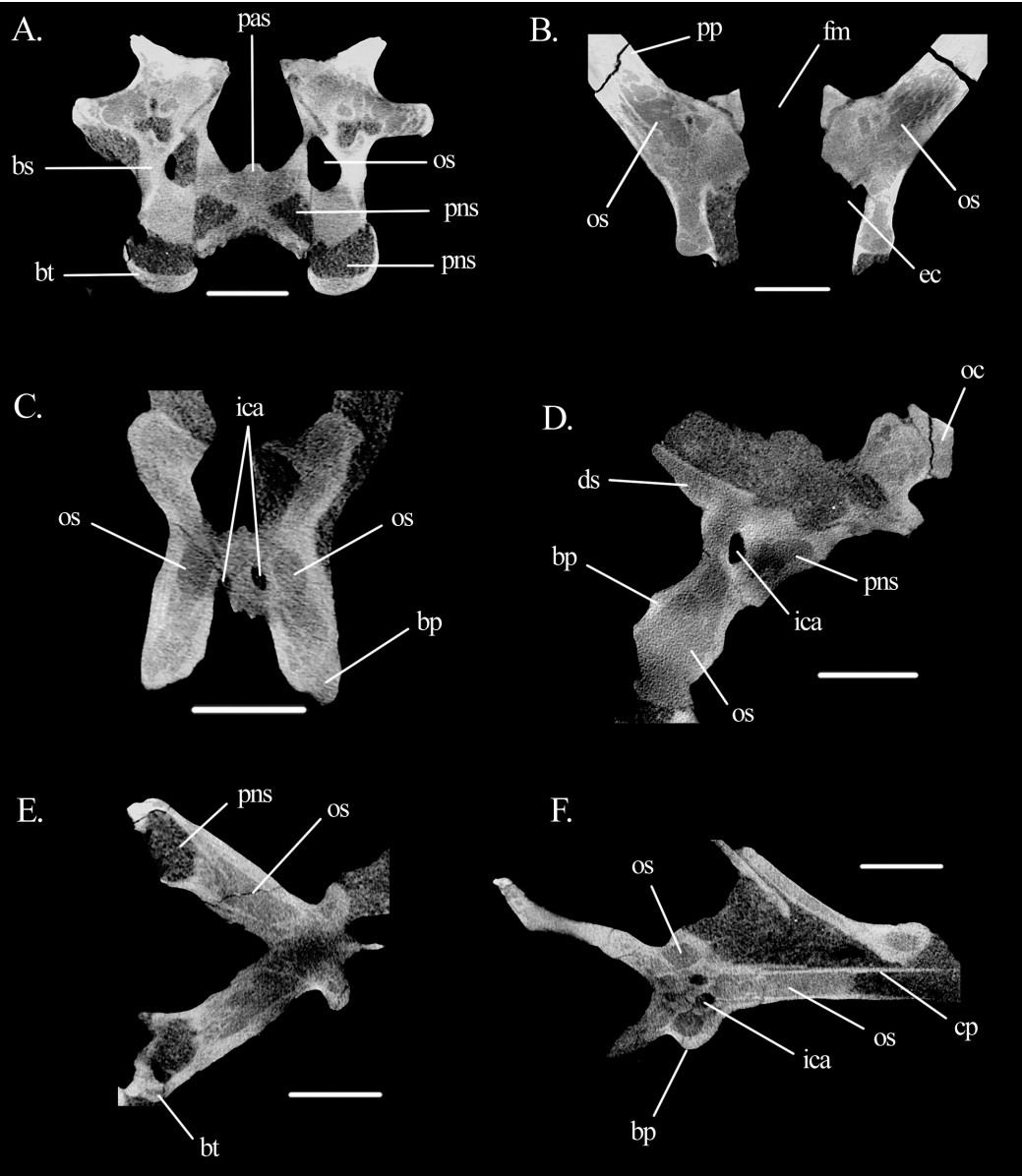

**Figure 9** **Slices showing open spaces and the pneumatic sinus in the braincase of *Mesosuchus browni*.**
Cross-sections of the braincase in (A) anterior view at the level of the basal tubera, (B) dorsal view at the level of the vestibule, (C) anterior view at the level of the basipterygoid processes, (D) lateral view at the level of the basipterygoid process, (E) dorsal view at the level of the basal tubera, and (F) dorsal view at the level of the cultriform process. Abbreviations: bp, basipterygoid process; bs, basisphenoid; bt, basal tuber; cp, cultriform process; ds, dorsum sellae; ec, endocranial cavity; fm, foramen magnum; ica, internal carotid artery; oc, occipital condyle; os, open space; pas, parasphenoid; pns, pneumatic sinus; pp, paroccipital process. Scale bars equal 4.5 mm in all, except (D) where it equals five mm.

## DISCUSSION

### General aspects

The dorsoventral height of the *Mesosuchus* braincase is greater than what has been interpreted in the literature (*Evans, 1986*; *Ezcurra, 2016*). It is more similar to *Trilophosaurus*

(*Gregory, 1945*) or *Erythrosuchus* (*Gower, 1997*), than to *Prolacerta* (*Evans, 1986*). The height in *Mesosuchus* is due to an increased dorso-ventral distance between the base of the basipterygoid processes and the basal tubera, and not so much by the distance between the tubera and the occipital condyle. The braincase of *Mesosuchus* is different from that of other, more derived rhynchosaurs, such as *Hyperodapedon* and *Stenaulorhynchus* (*Benton, 1983*), *Rhynchosaurus* (*Benton, 1990*) or *Teyumbaita* (*Montefeltro, Langer & Schultz, 2010*). In these, the parabasisphenoid is flat, so that the base of the basipterygoid processes and the basal tubera lie on the same level. In *Mesosuchus*, the posterior portion of the basisphenoid lies rather dosrsal to the anterior region. This character is unknown in *Howesia*, but the increased braincase height may be characteristic of basal rhynchosaurs, since *Eohyosaurus* shows a similar morphology (Fig. 10). The braincase of this rhynchosaur is poorly preserved, precluding a detailed description, but, when possible, it will be used here for comparison.

The crests on the prootic of *Mesosuchus* are homologous to the ones of *Euparkeria* (*Sobral et al., 2016*)—and in fact are present in most, if not all, diapsids. The crista prootica of *Mesosuchus* corresponds to "crest 1" of *Euparkeria*. It is well-developed in all archosauromorphs (*Ezcurra, 2016*), but not in stem-reptilians (*Gardner, Holliday & O'Keefe, 2010*; *Heaton, 1979*), where it does not extend as far ventrally or anteriorly. In crown archosaurs, on the other hand, the anteroventral extension of the crista may be extremely pronounced laterally, especially in sauropodomorphs (*Madsen, McIntosh & Berman, 1995*) and theropods (*Majungasaurus*—*Sampson & Witmer, 2007*). The second ("crest 2" in *Euparkeria*) has no official name and its presence in other archosauromorphs is obscure. The crista alaris corresponds to "crest 3" of *Euparkeria* and is present in several diapsids (*Oelrich, 1956*; *Sampson & Witmer, 2007*; *Sobral, Hipsley & Müller, 2012*). CT scans of both *Mesosuchus* and *Euparkeria* show that the crista alaris follows the outline of the anterior semicircular canal. In diapsids, including crown archosaurs, the smooth dorsal surface of the crista serves as the attachment site of the M. pseudotemporalis superficialis (*Oelrich, 1956*; *Sampson & Witmer, 2007*; *Sobral, Hipsley & Müller, 2012*). The rim may also have served as the attachment site of the prootic membrane, as in extant squamates (*Oelrich, 1956*).

The "fossae" on the opisthotic of *Mesosuchus* lateral to the foramen magnum were first noticed by *Ezcurra (2016*: 174), who stated the soft tissue correlation as "unknown" and thus did not identify them. Due to the topological correspondence to archosaurs (*Sobral, Hipsley & Müller, 2012*) and other reptiles (*Bruner, 1907*; *Dendy, 1909*), we regard them as representing the external openings of the vena capitis dorsalis. The foramina are present in the rhynchosaur *Howesia* (*Dilkes, 1995*), and perhaps also in *Teyumbaita* (Montefeltro, 2018, personal communication). Outside the archosaur crown, they are also found in archosauriforms such as *Fugusuchus* and *Garjainia prima* (*Gower & Sennikov, 1996*) and *G. madiba* (*Ezcurra, 2016*). Given the presence of an open space within the opisthotic (see below), no canal was found running within the bone. The vein must have run alongside the internal surface of the braincase and exited it in the area between the prootic, supraoccipital, parietal, and laterosphenoid (or the prootic membrane, if this bone was absent) as in other reptiles (*Bruner, 1907*; *Dendy, 1909*; *Sobral, Hipsley & Müller, 2012*).

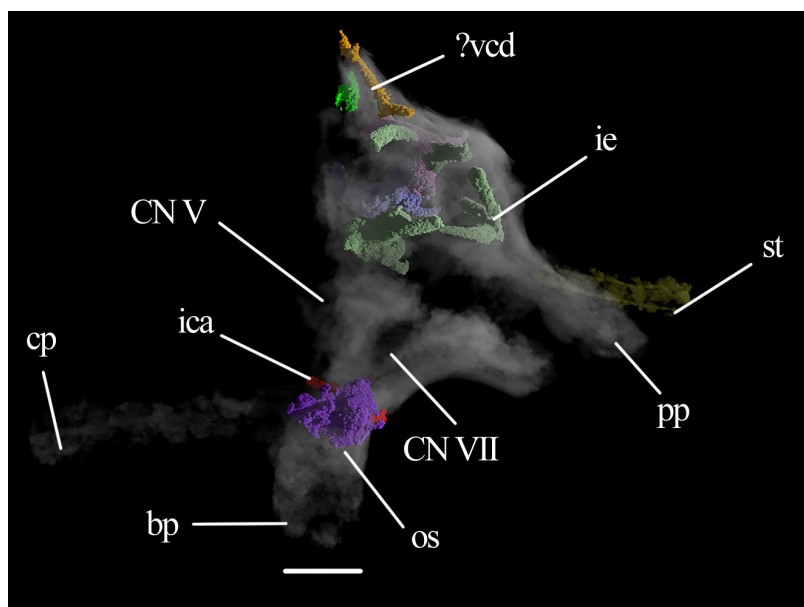

**Figure 10 Braincase of *Eohyosaurus*.** Braincase of the specimen SAM-PK-K10159 in left lateral view with several intraosseous structures volumized. Abbraviations: bp, basipterygoid process; cp, cultriform process; ica, internal carotid artery; ie, inner ear; os, open space; pp, paroccipital process; st, stapes; vcd, vena capitis dorsalis. Roman numerals refer to corresponding cranial nerves (CN). Scale bar equals four mm.

At the junction of the supraoccipital and prootic, there is a depressed and smoothened area that formed the posterior part of the internal foramen (Figs. 5A and 5B), like in *Euparkeria* (*Sobral et al., 2016*).

## Middle ear

The deeply recessed nature of the fenestra ovalis of *Mesosuchus* is similar to *Stenaulorhynchus* (*Benton, 1983*) and *Teyumbaita* (*Montefeltro, Langer & Schultz, 2010*), but contrasts with *Hyperodapedon* (*Benton, 1983*), and *Rhynchosaurus* (*Benton, 1990*). Outside Rhynchosauria, recessed fenestrae are found in the archosauriforms *Erythrosuchus* (*Gower, 1997*) and *Sarmatosuchus* (*Gower & Sennikov, 1997*), but also in phytosaurs (*Hungerbühler et al., 2012*). The implications of a recessed fenestra ovalis are not clear. In general, the degree of ossification the middle ear cavity and related elements are responsible for the upper limit of the hearing frequency (*Lombard & Hetherington, 1993*). At high frequencies, much of the energy of a soundwave is lost due to flexion of the system. Thus, ossification allows for a better transmission of higher frequency sounds. In aquatic environments, thickening of osseous structures of the middle ear and the presence of adjacent fat tissues seems to increase the efficiency of the transmission of high-frequency sounds (*Berta, Sumich & Kovacs, 2015*). Although *Mesosuchus* is obviously not an aquatic animal, some features of its middle ear may indicate similar functions. The deep location of the fenestra ovalis in the lateral wall of the otic capsule results in a significantly thicker, osseous border, which could facilitate sound conduction to the inner ear. Further, the occupation of this space with tissue lining and/or ligaments between the otic capsule and the footplate of

the stapes may enhance the direction of sounds waves into the capsule, in a manner similar to extant cetaceans (Berta et al., 2015). On the other hand, a stout stapes does not favor the conduction of high-frequency sounds. As the vibration of such a robust element does not occur easily, it hampers the transmission of high-frequencies.

In extant reptiles, the tympanic membrane attaches to the otic notch (=tympanic crest; otic conch) on the quadrate (*Wever, 1978*), and thus the presence or absence of this structure is commonly used to infer the presence or absence of a tympanic membrane in fossil taxa. Since derived rhynchosaurs lack this structure, *Benton (1983*, *1990)* inferred that *Rhynchosaurus* and *Hyperodapedon* also lacked a tympanic membrane and relied on low-frequency hearing. He hypothesized that intra-bone sound conduction could have been facilitated by the large hyoid elements, which would have been connected to the extrastapes, similar to the condition in *Sphenodon*. A robust hyoid is preserved on the ventral part of the skull of *Mesosuchus* (Fig. 2B), close to the left mandible. It is pictured and identified in figure 5B of *Dilkes (1998)*, but there is no mention to it in the text. Initially, this would point to a similar condition inferred to derived rhynchosaurs, but stating the presence or absence of a tympanic membrane in *Mesosuchus* is less obvious. The quadrate of *Mesosuchus* is concave (Fig. 2D), the left side much more so than the right, but a proper notch or conch is missing. This could indicate the lack of a tympanum, but recently other osteological correlates have been proposed to infer the presence, position, and extension of the tympanum in crocodyliforms (*Montefeltro, Andrade & Larsson, 2016*).

Following the proposed correlates, *Mesosuchus* could have had a meatal chamber, albeit more dorso-ventrally, instead of antero-posteriorly, extended. The chamber is not overhung by the squamosal, which is positioned more anteriorly in *Mesosuchus*, thus forming the anterior limit of the chamber together with the quadratojugal. The lateral flange of the quadrate has a concave surface so that the quadrate body forms a prominent ridge, which is also concave in lateral view (Figs. 11A and 11B). The surface is damaged on the left side, but on the right, it does not seem flat: a low, faint rim appears to delimit a shallow depression medially, which would correspond to the periotic fossa. Anterior to this rim, there are muscle scars. The tympanic membrane would have attached along the rim of the periotic fossa anteriorly and ventrally. Posteriorly and dorsally, it could have attached to soft tissue or cartilaginous structures. The posterior border of the meatal chamber is not bounded by bones, like in some basal crocodyliforms (*Montefeltro, Andrade & Larsson, 2016*). Likewise, *Mesosuchus* also seems to lack an otic buttress and other osteological structures posterior to it. However, in contrast to basal crocodyliforms, the dorsal border of the meatal chamber would not be delimited by bone due to a short, dorso-ventral expanded squamosal, as explained above. In this scenario, the dorsal ridge of the quadrate body could represent the otic incisure, which in this case would lack a bony dorsal component. These similarities seem to indicate that *Mesosuchus* possessed a tympanic membrane, although some caution is warned. A more thorough and broad analysis of the outer ear anatomy of stem-archosaurs is out of the scope of this work, and such homology tests can refute this in the future.

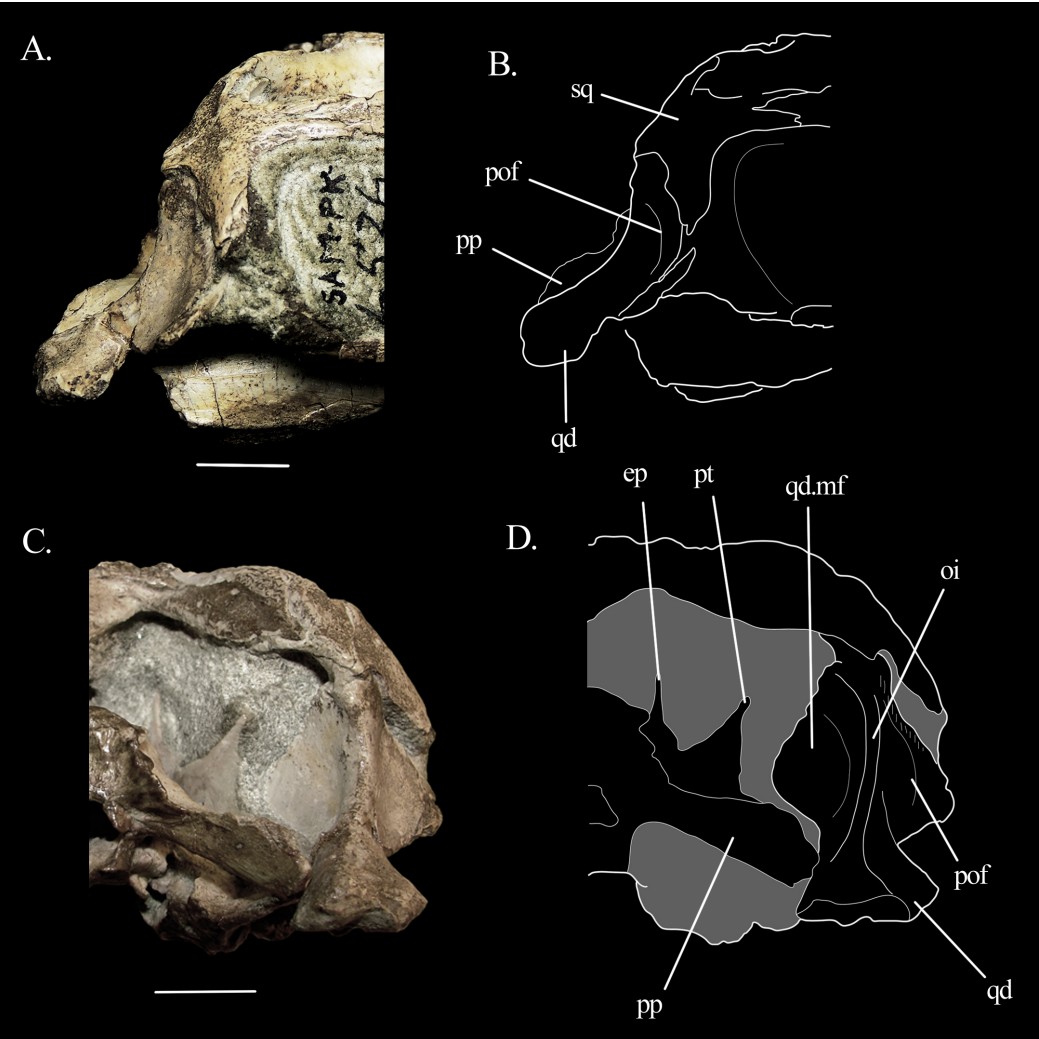

**Figure 11 Quadrate of *Mesosuchus*.** Details of the quadrate of *Mesosuchus browni*. Photographs of the posterior region of the skull in (A) right lateral view with (B) the corresponding line drawing and in (C) occipital view with (D) the corresponding line drawing. Abbreviations: ep, epipterygoid; mf, medial flange; oi, otic incisure; pof, periotic fossa; pp, paroccipital process; pt, pterygoid; qd, quadrate; sq, squamosal. Scale bars equal one cm.

The posteroventral inclination of the stapes in *Mesosuchus* (Fig. 8A) appears at first to indicate it articulated with the robust hyoid, but the braincase is strongly displaced. Thus, it is likely that the stapes did not extend so far ventrally, being then directed toward where the tympanic membrane would have been attached. The posteriorly directed stapes of *Eohyosaurus* seems to support that (Fig. 10). In the light of the current evidence, we do not think intra-bone conduction was the primary means of sound conduction in *Mesosuchus*. However, the recess around the fenestra ovalis and its dense borders, together with a stout stapes, may have facilitated the transmission of lower frequencies. This otic anatomy could indicate a trend towards a low-frequency hearing specialization started before the loss of the tympanic membrane. This is, however, not possible to test at the moment. While the quadrate of *Howesia* (*Dilkes, 1995*) and *Eohyosaurus*

(*Butler et al., 2015*) are very similar to the one of *Mesosuchus*, their braincases are either missing or very poorly preserved, respectively, precluding an assessment of the anatomy of their fenestra ovalis.

The braincase floor is slightly concave below the cochlea, but it did not excavate the dorsal surface of the basisphenoid, so that a cochlear recess is absent—contraty to *Euparkeria* (*Sobral et al., 2016*: 28; figure 3D). Therefore, both an unossified gap (*sensu Gower & Weber, 1998*; see *Sobral et al., 2016* for discussion) and a semilunar depression are absent in *Mesosuchus*. However, the borders of the original gap/depression are visible on the lateral surface of the basal tubera, ventral to the fenestra ovalis (Figs. 3E, 3F and 5C). A similar feature may be present in *Hyperodapedon* (*Benton, 1983*) and *Teyumbaita* (*Montefeltro, Langer & Schultz, 2010*), although it seems open in the latter. We are not able to conclusively refute the hypothesis of a connective tissue lining (*Evans, 1986*) nor of a cartilage coverage (*Gower & Weber, 1998*), although the latter is more probable given the endochondral process of ossification of the basisphenoid (*Bellairs & Kamal, 1981*). Such a cartilaginous growth plate has never been identified in any other fossil archosauromorph.

### Inner ear

Comparisons of the inner ear with other fossil diapsids outside crown groups is difficult because not much is yet available. They are tentative and must be taken with caution. Two important taxa to which information is available are the non-saurian diapsid *Youngina* (*Gardner, Holliday & O'Keefe, 2010*) and the archosauriform *Euparkeria* (*Sobral et al., 2016*). These will be used here. The semicircular canals of *Mesosuchus* are long and slender compared to the overall size of the vestibule, and the anterior one is slightly longer than the other two. This is similar to the condition found in *Euparkeria* (*Sobral et al., 2016*), but in sharp contrast with the one in *Youngina* (*Gardner, Holliday & O'Keefe, 2010*). Likewise, the large and deep floccular fossa of *Mesosuchus* is more similar to the fossa of *Euparkeria* than the one of *Youngina*. A large fossa is suggestive of a corresponding enlarged floccular lobe.

The semicircular canals and the flocular lobe are part of the vestibulo-ocular and vestibulocollic reflexes (*Witmer et al., 2008*). These circuities are related to sensing and coordination of movements of the head, eyes, and neck, and thus the semicircular canals and the floccular lobe are, at least indirectly, related to agility, locomotion, behavior, and ecology. The thickness, curvature, and angulation of the semicircular canals have all been used to infer these abilities (*Boistel et al., 2011*). Likewise, the size of the floccular fossa has also been used as a parameter for assessing gaze stabilization mechanisms (*Bronzati et al., 2017*; *Witmer et al., 2003*). However, it is not clear how these systems work in detail, and the role of the floccular lobe is particularly obscure (*Walsh et al., 2013*). The amount of neural tissue in a given structure is proportional to its processing capacities, so an enlarged lobe, and consequently an enlarged floccular fossa, is usually used as evidence for such refined mechanisms in fossil taxa (*Walsh et al., 2013*). However, the floccular lobe is not the only structure present in the fossa, which also contains the nodolus, the uvula, and several vascular elements. An enlarged fossa could be, at least

partially, due to the enlargement of these other structures, and not to the lobe only (*Walsh et al., 2013*), and a direct link between fossa volume and ecology has not been found for either extant birds or mammals (*Ferreira-Cardoso et al., 2017*; *Walsh et al., 2013*). Nonetheless, long and slender semicircular canals and an enlarged floccular lobe are usually suggestive of a refined mechanism for stabilizing head and eyes movements and such mechanisms are at some level related to navigation in complex, three-dimensional environments and/or to upright postures, which are usually associated (*Dudley & Yanoviak, 2011*). The herein described anatomy of the semicircular canals and of the floccular fossa for *Mesosuchus* suggests it did possess a somewhat refined mechanism. It is similar to *Euparkeria*, which has an upright posture and whose putative predatory behavior would require such refined mechanisms for gaze stabilization (*Sobral et al., 2016*). At the same time, it is in strong contrast with the anatomy of *Youngina* (*Gardner, Holliday & O'Keefe, 2010*) whose sprawling habits would be related to simpler, two-dimensional environments.

In contrast to *Mesosuchus*, derived rhynchosaurs were suggested to have had a partial sprawling posture. This was based on the anatomy of the post-cranial skeleton for *Hyperodapedon* (*Benton, 1983*) and *Rhynchosaurus* (*Benton, 1990*) and on biomechanical models for *Stenaulorhynchus* (*Kubo & Benton, 2007*). The inner ear anatomy is not available for these rhynchosaurs, but we can infer at least part of the structure of the semicircular canals from the outline of the crista alaris on the prootic, because it follows the general outline of the anterior semicircular canal. In *Hyperodapedon* (*Benton, 1983*) and *Rhynchosaurus* (*Benton, 1990*), the radius of curvature of the anterior semicircular canal seems rather modest compared to *Mesosuchus*, but *Stenaulorhynchus* (*Benton, 1983*) seems to have possessed an intermediate sized anterior canal. This indicates canals that are more modest in proportion than the ones of *Mesosuchus*. The modest-sized floccular fossa of *Hyperodapedon* (*Benton, 1983*) also seems to conform to that. On a first approach, the inner ear and floccular anatomy seems to support the posture estimations made for these derived rhynchosaurs. However, it is not yet possible to state whether more erect postures or more agile modes of locomotion could be characteristic of basal rhynchosaurs because data is lacking for these taxa. The inner ear of *Eohyosaurus* is partially preserved and could have been of help, but disarticulation of the braincase bones preclude its reconstruction (Fig. 10). Likewise, surveying the inner ears of more derived taxa could show us whether there is a trend for the loss in refinement of gaze stabilizing mechanisms.

The otic capsule of *Mesosuchus* is slightly more ossified medially than that of *Euparkeria* (*Sobral et al., 2016*). The postero-dorsal part of the vestibule is delimited by a medial extension of the opisthotic (Figs. 5D and 5E), which is absent in *Euparkeria*. This may be common to rhynchosaurs, since *Hyperodapedon* exhibits the same feature (*Benton, 1983*). Ventral to this medial extension, the ventral ramus of the opisthotic encircles the perilymphatic duct almost completely, forming a deep notch (Fig. 6E) that is in contrast to the shallow notch of *Euparkeria*. The anterior region of the vestibule is also marked by well-developed medial extensions of the prootic (Figs. 5D and 5E). The presence of another medial extension of the prootic, pierced by the CN VIII, is similar to *Euparkeria*, *contra Sobral et al. (2016)*—the foramen identified as CN VII in the latter authors' figure 3E is incorrect.

## Pneumatization

The presence of a pneumatic sinus is the most remarkable and unexpected feature of *Mesosuchus*, because no other stem-archosaur is recognized as possessing braincase pneumatization. A pneumatic sinus is defined by *Witmer (1990)* as the penetration of an air-filled chamber with communication to the external environment into the surrounding bone through an extension (diverticulum). It results in the formation of open spaces within the bones, which are connected to the air-filled chambers by several foramina. Thus, to identify a pneumatic sinus, it is not only necessary to find an open space in a given bone, but also the entrance foramen of the diverticulum and which air system originates it. In this sense, at present, only one system of open braincase spaces can confidently be classified as a pneumatic sinus in *Mesosuchus*, that is, that of the basal tubera. It occupies most of the basal tubera and the opening foramina are located at their medial border. Identifying which pneumatic system gave rise to this sinus is, however, less obvious.

One of the most comprehensive compilations of the braincase pneumatic system of fossil archosaurs is that of *Witmer & Ridgely (2009)*, where the authors describe all pneumatic sinuses of the tyrannosauroid braincase. However, there is no sinus system that seems to correspond exactly to the one of *Mesosuchus*. In theropods, the medial pharyngeal system is responsible for forming the subsellar (located ventral to the hypophyseal fossa and anterior to the basipterygoid processes) and the basisphenoid recesses (between the basipterygoid processes anteriorly and the basal tubera posteriorly). On the other hand, the subcondylar system forms the lateral and medial subcondylar recesses. The opening foramina of the latter are located on the ventral side of the basioccipital, with the recess pneumatizing the bone closer to the occipital condyle, whereas in the former the apertures are located on the ventral aspect of the otoccipital (=opisthotic + exoccipital), with the sinus occupying the crista tuberalis (=ventral ramus of the opisthotic).

On the other side of the archosaur tree, there is the highly pneumatic braincase of pseudosuchians, including extant crocodylians. Recent work, like that of *Dufeau & Witmer (2015)*, brought new light into the pneumatic anatomy and ontogeny of living taxa, but data on fossil ones remain scarce. Braincase pneumatic sinuses are so far unknown in non-crocodylomorph pseudosuchians. In the extant *Alligator mississipiensis* (*Dufeau & Witmer, 2015*), and in some early crocodylomorphs such as *Sphenosuchus* (*Walker, 1990*), there is a pneumatic space within the basal tubera, which in these taxa are formed only by the basioccipital (basioccipital recess in the former; basioccipital diverticulum in the latter). Also, the sinuses of these taxa do not correspond precisely to the one of *Mesosuchus*, since the foramina open in the anterior region of the tubera. Nonetheless, they are more similar than the pneumatic sinuses of theropods, and thus we propose the pharygotympanic system as likely having given rise to the pneumatic sinus in the basal tubera of *Mesosuchus*. This makes the openings in *Mesosuchus* the entrance foramina for the pharyngotympanic tubes, also known in the literature as the lateral (or true) Eustachian tubes (*Dufeau & Witmer, 2015*).

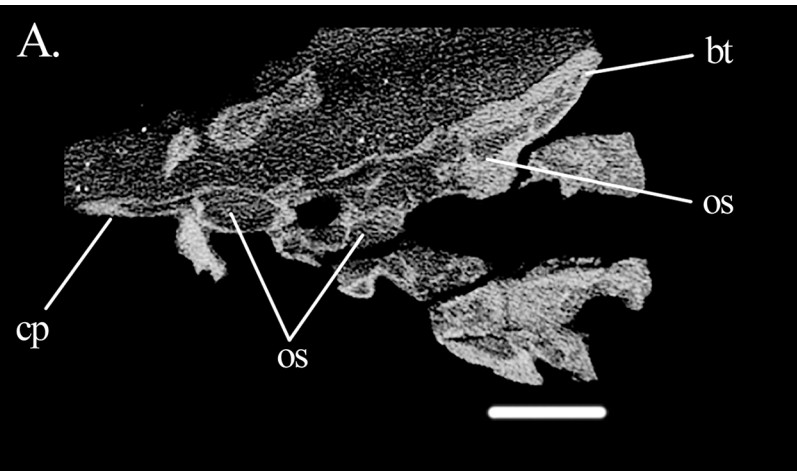

**Figure 12 Braincase of *Euparkeria*.** Cross-section of the braincase of the specimen UMZC T.692 of *Euparkeria* in dorsal view. Abbreviations: bt, basal tuber; cp, cultriform process; os, open space. Scale bar equals 3.5 mm.  

Classifying the open spaces in the parabasisphenoid of *Mesosuchus* as pneumatic sinuses is currently not possible. When comparing them with the sinuses of theropods, the spaces in *Mesosuchus* appear similar to the subsellar and basisphenoid recesses (*Witmer & Ridgely, 2009*). On the pseudosuchian side, the spaces in *Mesosuchus* differ from the basisphenoid diverticulum of the median pharyngeal system of extant crocodylians (*Dufeau & Witmer, 2015*), perhaps because the bone is much modified in these taxa. However, basal crocodylomorphs like *Sphenosuchus* (*Walker, 1990*) also have cavities in this region and, although they do not extend up to the prootic in *Mesosuchus*, they seem to correspond closely. Considering both cases, it would be expected for the opening foramen, or foramina, to be located in *Mesosuchus* somewhere on the basisphenoid between or around the basipterygoid processes; however, the only foramina present in this region are for the passage of the internal carotid artery. It is worth noting that similar open intra-bone spaces are found in the basipterygoid process of *Eohyosaurus* (Fig. 10) and in *Euparkeria* (*Sobral et al., 2016*), that is, in the cultriform and basipterygoid processes and in the main body of the parabasisphenoid of UMZC T.692 (Fig. 12). Opening foramina are lacking in these cases as well.

Also the space around the inner ear of *Mesosuchus* has its correspondents in archosaurs. It is similar to the caudal (=posterior) tympanic recess (PTR) of theropods (*Witmer & Ridgely, 2009*) and to parts of the occipital and intertympanic diverticula of alligators (*Dufeau & Witmer, 2015*). A similar sinus is not found in *Sphenosuchus* (*Walker, 1990*). The foramen of the PTR of theropods is located ventral to the crista prootica (=otosphenoidal crest), while in alligators it is found ventral to the fenestra pseudorotunda (incorrectly identified as the fenestra vestibularis in the figure 11 of *Dufeau & Witmer, 2015*). However, no foramina were found in *Mesosuchus*. Again, in *Euparkeria*, such a space is present in the paroccipital processes of SAM-PK-7696 (*Sobral et al., 2016*; fig. 3C therein), with no corresponding foramina.

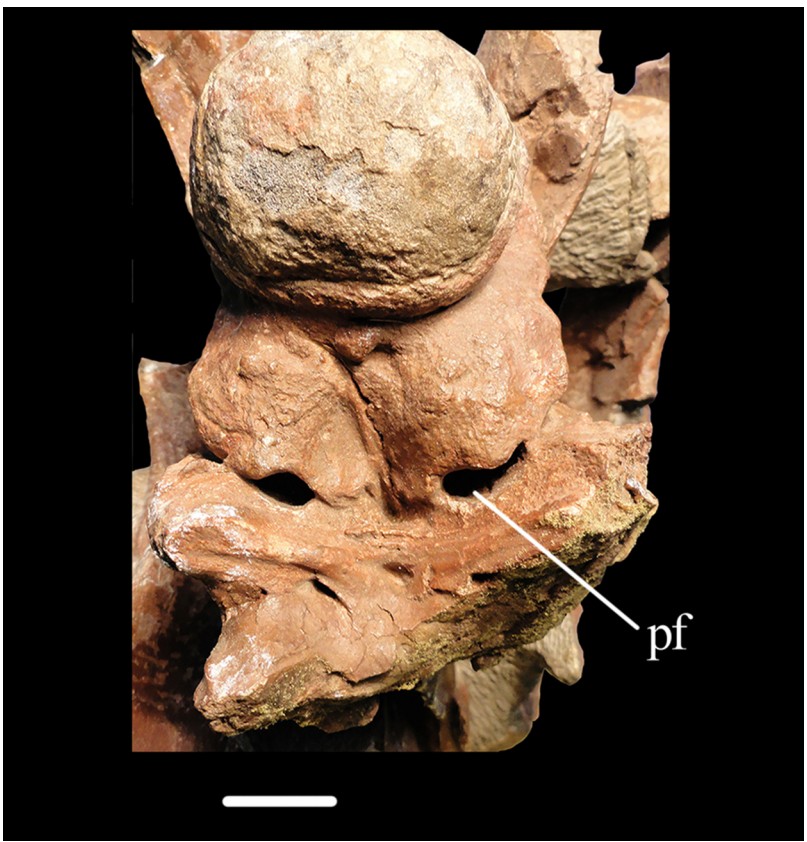

**Figure 13 Braincase of *Erythrosuchus*.** Detail of the braincase floor of the specimen BP/1/3893 of *Erythrosuchus africanus* in ventral view. Abbreviation: pf, pneumatic foramen. Scale bar equals 19 mm.

Even if only one pneumatic sinus can be confidently confirmed for *Mesosuchus*, this finding is noteworthy because, until now, pneumatic sinuses were thought to have been absent in all stem-archosaurs. However, it is possible that they may be more widespread in such taxa than previously acknowledged. Pneumatic sinuses in the basal tubera may also be present in other rhynchosaurs. For example, the depiction of *Dilkes (1995*; fig. 1B, D therein*)* of two small, symmetrical pits at the posteriormost margin of the ventral surface of the parabasisphenoid of *Howesia* may be indicative of the presence of pneumatic foramina, although these structures are not mentioned in the description. Likewise, *Benton (1983*: 632*)* mentioned "deep pits" on the basioccipital of *Hyperodapedon* posterior to the tubera, and illustrated them in his figure 10A. *Montefeltro, Langer & Schultz (2010*: 38*)* found similar structures in *Teyumbaita*. However, this structure seems to be absent in *Rhynchosaurus* (*Benton, 1990*) and in *Stenaulorhynchus* (*Benton, 1983*). Outside the group, foramina may be present in *Trilophosaurus* (*Gregory, 1945*) and in *Pamelaria* (*Sen, 2003*; fig. 4B therein). Likewise, two openings are present in the parabasisphenoid-basioccipital contact in specimen BP/1/3893 of *Erythrosuchus africanus* (Fig. 13), whereas no foramina have been depicted or mentioned by *Gower (1997)*. At first, these foramina may look like a preservational bias, as if the braincase floor had been antero-posteriorly

compressed, but given their similar position and shape, they could actually represent the external openings of a pneumatic sinus.

Inferring the presence of pneumatic sinuses in stem-archosaurs is a delicate issue, but the presence of a pneumatic sinus in *Mesosuchus* and, perhaps, also in other archosauromorph taxa does not necessarily imply that its presence is a plesiomorphic condition for archosaurs. Many archosauromorph groups lack it, including the most closely related terrestrial group of Archosauria, as represented by *Euparkeria* (*Sobral et al., 2016*). However, the discovery of a pneumatic sinus in *Mesosuchus* indicates that this feature is not exclusive of archosaurs, having evolved before the crown—and perhaps in different clades independently. Pneumatizaton of the axial skeleton has been suggested for *Erythrosuchus* (*Gower, 2001*), but *Mesosuchus* is the first documented case of pneumatization in the skull. Braincase pneumatization is found in both extant archosaur clades. In sauropodomorph and theropod dinosaurs, it could be primarily related to weight reduction, but in crocodylians it may be connected to directional hearing and to the detection of low-frequency sounds (*Dufeau & Witmer, 2015*; *Grigg & Kirshner, 2015*). Whether pneumatization in early archosauromorphs is related to hearing or to weight reduction of the skull remains to be tested. The discovery of a pneumatic sinus in *Mesosuchus* shows that the process of braincase pneumatization is more complex than previously thought.

## CONCLUSION

The redescription of the braincase of *M. browni* presented here contributes to our knowledge not only on the braincase anatomy of basal rhynchosaurs, but also of stem-archosaurs in general. The braincase of *Mesosuchus* is dorsoventrally taller than previously acknowledged (*Evans, 1986*; *Ezcurra, 2016*). The morphology of the anterior region of the prootic, where the trigeminal foramen is located, may indicate the presence of a laterosphenoid, although such an element could not be identified with certainty. Previous assessments on the paleoecology of *Mesosuchus* (*Dilkes, 1998*) depicted the taxon as a semi-erect animal similar to more derived rhynchosaurs (*Benton, 1983*; *1990*). However, the analysis of the inner ear of *Mesosuchus* shows a long anterior semicircular canal and an enlarged floccular fossa, indicating this taxon could have had a more agile mode of locomotion. Previous studies suggested that derived rhynchosaurs were not capable of detection of high-frequency sounds, based on the lack of a tympanic membrane and the presence of robust hyoid elements (*Benton, 1983*, *1990*). The presence of a robust hyoid, and the stout stapes seems to indicate *Mesosuchus* also had some specialization towards detection of low-frequency sounds. However, the recessed nature of the fenestra ovalis, with dense bony rims and the presence of a tympanic membrane, must still have favored higher frequencies. The most striking feature of the *Mesosuchus* braincase is the presence of a pneumatic sinus in the basal tubera. The opening foramina of this sinus have been previously identified as pits in other rhynchosaurs (*Benton, 1983*; *Dilkes, 1995*; *Montefeltro, Langer & Schultz, 2010*), indicating the potential presence of such structures in other species of the group. The presence of similar pits at the contact between the parabasisphenoid and basioccipital in stem-archosaurs such as *Erythrosuchus africanus*

also suggests that a pneumatic sinus may have been present in the braincase floor of clades other than rhynchosaurs.

## INSTITUTIONAL ABBREVIATIONS

**SAM**    Iziko South African Museum
**UMZC**    Cambridge University Museum of Zoology
**BP**    Bernard Price Institute for Palaeontological Research.

## ACKNOWLEDGEMENTS

We would like to thank Sheena Skal (South African Museum) for loan of the specimen, Jason Pardo (University of Alberta) for discussion, and Richard Butler (University of Birmingham) for the CT data on *Eohyosaurus*. We are also thankful to Felipe Montefeltro (University of the State of São Paulo), David Dilkes (University of Wisconsin), and Lawrence Witmer (Ohio University) for revising this manuscript, which helped significantly increase its quality.

### Funding

This project was supported by the DAAD and CAPES program with funding provided by CAPES (BEX 3474/09-7) to Gabriela Sobral. The funders had no role in study design, data collection and analysis, decision to publish, or preparation of the manuscript.

### Grant Disclosure

The following grant information was disclosed by the authors:
DAAD and CAPES program with funding provided by CAPES: BEX 3474/09-7.

### Competing Interests

The authors declare that they have no competing interests.

### Author Contributions

- Gabriela Sobral conceived and designed the experiments, performed the experiments, analyzed the data, prepared figures and/or tables, authored or reviewed drafts of the paper, approved the final draft.
- Johannes Müller conceived and designed the experiments, analyzed the data, contributed reagents/materials/analysis tools, authored or reviewed drafts of the paper, approved the final draft.

### Data Availability

Both CT scans are stored in the public digital collection of the Museum für Naturkunde Berlin and they can be accessed through the µCT lab (mikroctlabor@mfn-berlin.de) or, alternatively, through the authors. Additionally, the 3D model of the braincase

of Mesosuchus can be found online under the address sketchfab.com/models/
d332dbe1e6f8452ea8314d64989ddda7.

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
