# Peer review of "The braincase of Mesosuchus browni (Reptilia, Archosauromorpha) with information on the inner ear and description of a pneumatic sinus"

_PeerJ, doi:10.7717/peerj.6798_

## Round 0.1 · original submission · Major Revisions

Dear Gabriela,

I have now received three reviews of your paper submitted to PeerJ. The reviewers recommend a number of improvements, which should be addressed before re-submission. Besides general issues regarding some of your interpretations (e.g. pneumaticity), they request that:

-you improve your figures;
-you make your CT data available for review.

As per PeerJ policies (https://peerj.com/about/policies-and-procedures/#data-materials-sharing), all the raw data have to be made available in a permanent public repository prior to formal acceptance.

Please, do it before submitting the revised version of your work so as to make it easier for the reviewers to provide an adequate evaluation of your work in the second round of reviews.

Please, together with your unmarked revised manuscript, provide a marked-up copy as well as a document explaining how you have addressed each of the points raised by the reviewers.

Thank you for your attention.
Best regards,
Fabien

·

Basic reporting

The MS provides new anatomical information on the braincase of the early diverging rhynchosaur Mesosuchus browni based on the mCT data from the superb skull SAM-PK-6536. The most interesting result is the identification of pneumatic sinuses reported for the first time in this group. Such findings have important implications for the evolution of the pneumatic systems in Archosauromorpha. I detected which I consider problems that should be addressed before final publication.
I am not a native speaker, and I did not conduct a revision of the language. However, as far as I can tell, the MS employ proper scientific writing. The MS presents adequate background and a sufficient introduction to contextualize the new finding in the broader field of knowledge. In general, the relevant literature is appropriately referenced.

Experimental design

Apart from the lack of adequate comparison to other rhycnhosaurs and the pneumatic sinuses and ear of crocodylians, the investigation was conducted rigorously (see below).

Validity of the findings

One central aspect that needs improvements is the comparison to other taxa in the discussion section. The comparison among rhynchosaur is basically restricted to information provided by Benton (1983). There is no adequate comparison to Rhynchosaurus articeps, Stenaulorhycnhus, Hyperodapedon and Teyumbaita. The comparison to the taxon Eohyosaurus is limited yet the MS provide the reconstruction of its braincase.
The same lack of comparison is detected regarding the pneumatic sinuses and ear. The MS includes comparison to theropods but not to crocodylians. We have a fairly amount of information available on the pneumatic sinuses and ear of crocodylians (e.g. Tahara & Larsson 2011, Dufeau & Witmer 2015). Such comparison would greatly strengths the importance of the findings.
Given the position of Rhynchosauria outside crown-Archosauria, the comparison to lizards would also be informative.
In addition, the authors compare the new findings to Scaphonyx. However, Scaphonyx is considered a nomen dubium. The authors should provide the rationale behind the comparison to Scaphonyx.

Additional comments

One reservation I have about the paper is the quality of the line drawings. In the present form, the line drawings do not help on the evaluation of the anatomical information provided in the description. I recommend improving the line drawings. I also suggest some features to be highlighted in the figures (foramina for the cerebral branch of the internal carotid arteries, “crests 1-3”, recessus stapedialis, “two small fossae on the opisthotic”, perilymphatic notch, obliterated external foramina), and the addition of a figure of the ventral view of braincase.
All figures lack scale. Please, include scale to the figures.
Line 43. Previous papers also had referred rhynchosaurs as “pig-like cretatures” but it does not deliver the true picture about the group diversity. The group also includes tiny taxa such as Rhynchosaurus articeps with the approximated size of an agouti, and Brasinorhynchus which probably exceeded the size of a tapir. Although, it does not have a real impact on the scientific merit of this MS, it is an oversimplified view of rhynchosaur diversity.
Line 44. Mesosuchus does not present tooth plate nor beak-like premaxillae.
Line 48. Actually, rhynchosaurs became important faunistic components worldwide during Mid-Late Triassic (Schultz et al. 2016, Ezcurra et al. 2016).
Line 49. At least Teyumbaita crossed the Carnian-Norian boundary (Butler et al. 2015, Schultz et al. 2016, Montefeltro et al. 2010, Ezcurra et al. 2016).
Line 87. Please provide the full spelling of the institutional abbreviation.
Line 94-97. I am not sure if I understood the possible divergence between the Author’s opinion and Ezcurra’s opinion on the height of the braincase. In my opinion it all depends on the real articulation of the braincase to the skull. I suggest the author to elaborate more this possible divergence of interpretation.
Line 266. I failed to recognize the remarkable differences proposed between the braincase of Stenaulorhynchus and Mesosuchus. Please, elaborate it.
Line 287. I suggest to elaborate more on the reasons of the proposition of Mesosuchus hearing adapted to low-frequencies sounds.
Line 291. I am interested on the evidences to suggest that rhynchosaurs did not have tympanic membrane.
Line 321. I would like to see more evidences for the statement: “It is safe to postulate that Mesosuchus had an even more erect posture and was more active than both rhynchosaurs”. I failed to see the connection to the skeletal reconstruction from Dilkes (1998).
Line 366. Montefeltro et al. 2010 described a distinct depression of the exoccipital in Teyumbaita, which have topographical correspondence to the small fossae on the opisthotic.
Caption of figure 1. Specimen number is wrong.
Caption of figure 8. It is missing the PNS.

·

Basic reporting

No comment.

Experimental design

No comment.

Validity of the findings

No comment.

Additional comments

General Comments: This paper presents a significant advance in our knowledge of an important Middle Triassic archosauromorph represented by specimens that are wonderfully amenable to recent advances in non-invasive imaging such as micro CT scanning. The unexpected presence of pneumatic sinuses in the braincase and details of the inner ear hold great promise to inform phylogenetic hypotheses and enhance our understanding of the biology of Triassic reptiles.

Specific Comments:
Abstract. It’s common to describe rhynchosaurs as pig-like, but this really only applies to the later genera. Mesosuchus has more of a lizard-like body shape than that of a pig.

Introduction.
Line 45. Herbivory was likely true for later rhynchosaurs (perhaps including Howesia), but there is little to suggest such a diet for Mesosuchus. Mesosuchus seems more like an insectivore.
Line 46. The earliest record for rhynchosaurs is the Early Triassic, but the presence of the archosauriform Archosaurus in the Late Permian pulls the lineage back to an origin in the Late Permian.

Description.
Lines 95-97. Yes, the braincase is certainly not transverse. The dorsoventral orientation is also evident in Figure 2C & D of Dilkes (1998).
Lines 129-131. The description of the dorsum sellae in Dilkes (1998) was based on SAM-PK-6536. Although the middle region is not completely present, it was evidently a tall structure.
Lines 132-133. The median ridge described in this line is labeled in Figure 3B, not 3A.
Lines137-142. There is confusion here due to my poor writing regarding the crista ventrolateralis. As I understand it, the term should have been used to describe the ridge between the basal tuber and basipterygoid process with a posterior extension that contacts the basal tuber portion of the basioccipital. The crista ventrolateralis is a part of the amniote braincase. Do the authors identity it in Mesosuchus?
Lines 15-151. Can the two crests “crest 1” and “crest 2” be labeled in a figure?
Lines 152-154. Would it be better to describe it as being initially anterodorsal to the foramen for CNVII and reaches the anteroventral side of this foramen as it extends ventrally?
Line 193. “…external foramina of the of the vena capitis dorsalis…” should be changed to remove one of “of the”.
Line 196. “Ventral do it…” should be “Ventral to it…”
Lines 204-206. Dilkes (1998) did describe a pair of foramina for CNXII in a deep fossa on the lateral side of the exoccipital.
Lines 210-212. Choice of words to describe the shape of the stapes has clear implications for impressions of function. Calling it stout automatically implies in the mind of the reader poor ability to conduct high frequency sounds (a function presumably restricted to a slender stapes). Why is this stapes stout rather than slender? The width of the shaft does not appear to be markedly different from the width of the head described by the authors as small. What anatomical criteria are the authors using to distinguish between stout and slender?
Line 290. “…although no mention to it…” should be changed to “…although no mention of it…”.

Figures.
Figure 2. The figure has bo.rd that is clearly a combination of labels bo and rd. Should have bo.rd as an abbreviation in the figure caption. Similarly, st.bo.ps should be one of the abbreviations.
Figure 3. Label on figure is cr3, but figure caption has c3.


References.
The following papers listed in the References could not be located in the body of the paper or figure captions:
Brusatte et al. (2016)
Cordorniú et al. (2016)
Dzik (2003)
Gow (1975)
Hungerbühler et al. (2012
Kubo and Benton (2007
Marugaán-Lobón et al. (2013)
Nesbitt (2007)
Sereno et al. (2007
Thewissen and Nummela (2008)
Witmer et al (2003)

Thank you for the opportunity to review this manuscript. It's nice to see another example of how new technology can be used to correct errors in past papers and provide significant new anatomical information.

·

Basic reporting

This is a generally very well done manuscript that does a good job of describing and interpreting the morphology of the braincase of the basal rhynchosaur Mesosuchus. The structure of the manuscript is fine. The English is also generally very good, but could benefit from a close read by an editorial eye.

As for the figures:
1. The Introduction needs a cladogram with all of the taxa mentioned in the manuscript, as well as the nodes, if only so the reader can get a sense of the phylogenetic framework. Many taxa are mentioned, which is good, but we need to know how they all fit together, even if just plotted on someone else’s tree. This is required in my view.
2. The inner ear isn't reconstructed in isolation and compared with other relevant species, and such a figure would seem to be not only possible but absolutely necessary. It’s almost promised by the manuscript’s title.
3. The photos in Figures 2 and 3 are very hard to interpret because the lighting is nonstandard and many important features are in dark shadow. The interpretive line drawings also aren't very helpful and they're not extensively labeled.
4. The 3D volume renders of the skull are quite muddy and dark. It's hard to see detail. Moreover, considerable detail is lost in the dark, raking shadows, which is a persistent problem with rendering in VG Studio Max.
5. In general the figures aren't labeled enough.
6. Figures with slices (e.g., Figures 6 & 8) should have an inset of the braincase showing where the slice comes from. Otherwise, they're very hard to orient, made worse by the fact that they're not labeled very much, and again are very dark and muddy, especially in Figure 8.
7. It’d be very nice to have comparative drawings of relevant taxa showing key features compared to Mesosuchus. It would make the manuscript much more useful.

Experimental design

A CT-based analysis of the braincase of Mesosuchus is highly appropriate and welcomed given the basal position of Mesosuchus within the rhynchosaur clade as well as the basal position of rhynchosaurs within Archosauromorpha. The authors are experts—especially the senior author—in basal archosauromorph braincase structure, and thus this project is on a sound footing. This kind of work hasn’t been before for this clade so it’s absolutely needed.

A major concern, however, is that the authors do not seem to have taken any steps to make the CT scan data available (e.g., on Dryad or MorphoSource). In fact, it was hard for me to do an adequate review of the manuscript without being able to see the scan data. Making the data available on an open-data portal is now expected in that it addresses the need for repeatability and replication. The absence of providing the data actually seems to be a violation of PeerJ policy, although I’m no expert on PeerJ policy.

Validity of the findings

The morphological findings are generally fine although I’ll point out some concerns under General Comments. Again, the figures need some attention. The Conclusions section is generally fine in terms of summarizing some of the major outcomes of the work, although a lot of it pertains to the pneumaticity interpretations about which I’m less confident (see my General Comments).

Also, there’s one statement in the Conclusions—lines 404-406: "The inclination of the lateral semicircular canal indicates Mesosuchus held its head horizontally in relation to the rest of body, also in agreement with previous suggestions."—which is the first time that this point is made. It shouldn’t appear for the first time in "Conclusions," but rather should be presented in more detail in the body of the manuscript. Moreover the statements requires references: "...in agreement with previous suggestions"…whose previous suggestions?

Additional comments

I really liked this manuscript, as it presents very important information about an important species. I think the figures really need to be improved. The manuscript relies extensively on CT-based imagery, which in itself can be okay, but the images here need to be—and I think could be—much better.

Some more specific points are made here:
1. Lines 149-163, 272-280: The sections (there are two) on the braincase crests need some work. First, they’re not well illustrated. Only crest 3 is labeled (Fig. 3) and Figures 4 and 5 would also be possible places to label the crests. Why call “crest 1” anything other than crista prootica (or my preferred term [see below] otosphenoidal crest). This synonymy is not revealed until line 272 and not in the section on the prootic. To be honest, I’m not sure what’s going on with these crests—I wasn’t sure in recent Euparkeria article and I’m not sure here either. Better figures might help, but also seeking really clarity in the text. Function might help, too, since soft tissues are making these crests.
[Aside: It might be useful to note that the “crista prootica” has also been referred to as the “crista otosphenoidalis”—and not just by me. I didn’t invent the term, either, but I prefer it because in many species (including Mesosuchus, apparently), it’s not restricted to the prootic bone but actually runs from the paroccipital process (opisthotic), across the prootic bone, and down to the basipterygoid process of the basisphenoid bone…that is, it’s literally a crest that runs from the otic region to the sphenoid region. It’s an extremely consistent feature that functionally separates the tympanic domain from the adductor and orbital domains.]

2. Line 214: What does "it" refer to? Footplate or fenestra ovalis? Also, how can you not confirm whether the footplate fills the FO. It obviously doesn't in Figure 6

3. Line 218: "The fenestra ovalis is very small..." It looks huge in Fig. 6B.

4. The identification of pneumaticity is a pretty big deal here because it’s so unexpected. The authors themselves seem very tentative at times, and I think that’s fair. I’m not fully convinced either. I’ve devoted a lot of thought (and a lot of ink) to pneumaticity over the years. An important point I’d like to make at the outset is that I have no a priori bias against pneumaticity in Mesosuchus. It’s totally possible, doesn’t conflict with any pet ideas of mine, and would be really interesting. So let’s dig in… Lines 241-242: "...the braincase of Mesosuchus is trabeculate, which makes it difficult to assess the real extension of the system." Yes, it can be very difficult. Lots of extant squamates have a quite open trabecular structure of the skull bones...but they're not pneumatic. The key attribute for pneumaticity is a pneumatic aperture (foramen) that opens to (communicates with) a known air-filled space, like the middle ear or pharynx. Can you identify the aperture in all cases? If not, the case for pneumaticity is weaker. Lines 251-252: "Its external foramina were could not be identified..." Delete "were." Also, it seems very unlikely that the foramina on the occiput lateral to the foramen magnum are the pneumatic foramina. What would be the air-filled space providing the source of the diverticulum? The tympanic cavity isn't there. More on this point below. Lines 343-350: The identification of foramina leading into open spaces is critical for the inference of pneumaticity. The authors report finding foramina in lines 343-344, but then in the next paragraph they say that they don't have foramina. Lines 351-360. The authors identify a putative caudal tympanic recess but lack foramina connecting to the middle ear, which would be the source of the air. I agree that the "obliterated foramina" (are they really obliterated?) are for the external occipital vein, which is highly conserved, but it seems odd to then call it a pneumatic foramen. First, if it's "obliterated" then why would there be a pneumatic space, since in extant amniotes patency of the pneumatic foramen is required for maintenance of the pneumatic space. Second, what would be the source of air passing through the "obliterated foramen"? The choices would be middle ear and lung but both of these seem very unlikely given their distance. Lines 366-368: Fossae on the occiput seem like very weak criteria for pneumaticity. Again, how would a diverticulum of the tympanic cavity get there? Also, what about the cervical musculature? Certainly neck muscles attach to the occiput in this area and could make these fossae. Such depressions for muscular attachment are common. Also, using the "obliterated foramina" of Mesouchus as a basis to infer pneumaticity in other taxa seems unwarranted.

5. Lines 288-289: How does the absence of a quadrate conch and having a robust hyoid relate to low-frequency hearing? References?

6. Line 291: How do we know that "rhynchosaurids lack a tympanic membrane.”

7. Inner ear. The inner ear isn't reconstructed in isolation and compared with other relevant species, and such a figure would seem to be possible. Lines 308-309: "also suggestive of an active animal with a more upright posture." What is the basis of this statement? No reference is provided. Such a statement about activity patterns and posture is quite bold given that the link between inner ear structure and activity and posture have been controversial at best. At a fundamental mechanistic level, the SCC and cerebellar flocculus are associated with sensing head and body movements and coordinating eye movements. Any connection to activity and posture must flow from the neurology, which is indirect at best in this case. Lines 313-321: Comparison to Protoceratops—and certainly the inner ear of Protoceratops—are of doubtful utility given the vast difference in phylogenetic history.

---

## Round 0.2 · Minor Revisions

Dear Gabriela,

The referees who re-reviewed your work find that it has been significantly improved. They do, however, still have some suggestions for improvement. In particular:

-Some photos suffer from poor lighting and/or are partially out of focus: please, make every effort to have them retaken (especially Fig. 2).
-Make sure that the raw CT data are reposited at the MfN Berlin (not on your personal Google Drive) and that specific instructions for access are given in the manuscript (this is not the case in the current version).

Please, together with your unmarked revised manuscript, provide a marked-up copy as well as a document explaining how you have addressed each of the points raised by the referees in this second round of reviews.

Best regards,
Fabien

·

Basic reporting

The authors present an improved version of the MS. In the new version, the comparisons are better conducted and most of my concerns had been addressed. However, the MS still contains what I consider minor problems that should be addressed before final publication.

Experimental design

No comment

Validity of the findings

I still think that comparison to lizards would greatly improve the reliability of the generalizations made about the evolution of the pneumatization. However, it is fair if they prefer to restrict the scope to Archosauromorpha.

I failed to see in which manner the basipterygoid process of Mesosuchus is different from Stenaulorhynchus and Rhynchosaurus (lns 262-263). The more developed basipterygoid process is recognized as a plesiomorphy of Rhynchosauria. Only Hyperodapedon and Teyumbaita present a shorter process (see Schultz et al. 2016, Langer et al. 2017 character 52). Please clarify.

Other osteological correlated are proposed for inferring the presence and extension of the Tympanic Membrane in crocodylian lineage (see Montefeltro et al. 2016).

Additional comments

Garjainia prima not in italic, please change (ln 283).

The pictures of the braincase in figures 3 and 4 are too bright in the pdf version of the revised MS. If this is intended for the final version, it should be re-worked.

·

Basic reporting

This new version of the manuscript is much improved. I’m grateful to the authors for providing access to the CT scan data because it was indispensable and made possible an adequate assessment of the morphology. As before, the manuscript is well written, and the basic structure of the manuscript is very sound.

Thank you for adding the new Figure 1 in that the cladogram will be very helpful for those less familiar with the group…like I was with the first review…I’ve since dug into the rhynchosaur literature a lot more than I did the first time. I also appreciate the authors’ adding more labels to the line drawings.

However, the authors chose not to change the figures very much and some of the changes are now for the worse. I was hoping for better versions of the photographs in Figs 3 and 4 but it looks like they just decreased the contrast in a very coarse way such that they now just look washed out. It didn’t help and they now look worse. If the first versions are the best that can be done then so be it, although if photos can’t be reshot then maybe some skillful application of Photoshop might be able to make the photos more useful. If not, then the first versions are preferable.

I appreciate that the authors generated new versions of the 3D volumetric renders of the braincase in Figs 3–6, but these also seem blown out and with no shadows or contrast; they look like flash photographs of white plaster casts. Once again, I think the first versions are preferable. Having worked with the dataset now myself, I understand that it’s a little tricky, but maybe they could try again. Likewise, the slices in Fig 7 are a little blown out whereas they’re pretty nice in Fig 9. I was disappointed that the authors decided not to take my recommendation to have a separate inner ear figure. Figure 8 is nice, but it’d be helpful to see the isolated labyrinth in lateral and dorsal view.

I don’t mean to be overly picky on the figures but this work is really nicely done and important, and it would be great if the figures matched the manuscript in quality.

Experimental design

Experimental design is fine

Validity of the findings

Validity of findings is fine although I make some points in General Comments.

Additional comments

p. 10, line 107: "opening foramen of the pneumatic sinus" Or is this the Eustachian (pharyngotympanic) tube opening?

p. 11, line 131: "median pharyngeal recess" This doesn't seem like much of a recess. It's just the normal concavity. It true that it’s rather arbitrary to decide whether a normal fossa is deep enough to be called a “recess.”

p. 12, lines 153-155: "This third crest runs from a point anterior to the ventral bending of the crista prootica sharply anterodorsally to the anterior margin of the prootic. It terminates dorsal to the foramen of CN V. The lateral surface of the prootic dorsal to this third crest is smoothly depressed." This crest and the fossa behind it are widely distributed (Desmatosuchus, Allosaurus, others), where it's likely the attachment for musculature.

p. 12, lines 161-165: "The foramen of CN VII is damaged on the left side. On the right, it is positioned posterior to the crista prootica, so that it is visible in lateral view of the braincase. A shallow sulcus runs postero-dorsally from the foramen of CN VII along the ventral surface of the crista prootica and represents the hyomandibular branch of the facial nerve." I don't think that the facial nerve foramen is correctly identified. I think it's more dorsal, tucked under the crista prootica rather than being the large opening labeled in the figures. The more dorsal opening is visible in the CT data on both sides. Moreover this is precisely where the facial nerve foramen is located in a diversity of archosaurs. The hyomandibular and palatine branches of the palatine nerves travel just under (ventral to) the crista prootica. I'm not sure that the feature identified as the facial nerve foramen is truly an aperture, but may just be very thin bone. It also would be a giant aperture for such a small nerve.

p. 12, lines 170-172: "Posterior to the ventral part of this projection, and dorsal to the foramen of CN VII, another small projection bearing a foramen can be seen. The foramen is identified here as the entrance of the posterior branch of the auditory nerve (CN VIII; Fig. 5D, E)." I'm not sure that there's really a "foramen" there as much as a large unossified space. I looked pretty hard at the CT data and I’m not convinced of a discrete foramen, which is consistent with poor ossification of the medial wall of the inner ear which is common in this part of the phylogenetic tree.

p. 13, line 183: "two small fossae on the opisthotic…" I only see the one on the left. There's nothing on the right that I'd call a fossa. On neither side is there anything that I'd regard as an opening for either vasculature or pneumaticity. They're labeled "pf" in Figure 3, but that seems without adequate basis. I also see no basis for regarding any apertures as "obliterated" because I see no evidence that there ever was an opening.

p. 15: Pneumatic sinuses: the ventral opening may indeed be the pharyngotympanic (Eustachian tube) opening. The space it leads into is completely confluent with the inner ear, which makes no sense for pneumaticity (which is middle ear). The challenge here is that the area may have been really poorly unossified. There's also the unossified gap between the basioccipital and (para)basisphenoid contributions to the basal tubera. That gap within the basal tubera (often a very roughened area) is extremely common among basal archosaurs (even some dinosaurs). The authors refer to this gap as a pneumatic sinus in Figure 9A, but I think it's just the gap between the bones. The fact that this gap is continuous with the sinus raises the possibility that some of the "sinus" could be simply unossified areas.

p. 16-17: "Two other systems of open braincase spaces". I agree that there are some rather compelling open spaces with smooth walls that look like pneumatic spaces. However, they indeed lack foramina opening into an air-filled space, and are continuous with clearly cancellous or trabeculated regions. They may simply be marrow spaces. Lots of extant lizards have similar large, smooth-walled diploic cavities. The case for pneumaticity is not strong but I understand the authors' willingness to consider the case. Their treatment of it is now sufficiently measured. Well done.

p. 19, lines 314-315: "The quadrate of Mesosuchus also lacks a notch, indicating a tympanic membrane was absent in this taxon." I'm not sure that I fully agree. There seems to be a pretty reasonable tympanic crest running the length of the quadrate laterally and even a gentle "conch" caudally. Even if more derived rhynchosaurs lack the tympanum, Mesosuchus is so basal that it may have retained a tympanum.

p. 20, lines 337-338: "the unossified gap represents the ventralmost portion of the lagenar recess (Sobral et al., 2016:28; figure 3D), but this is clearly not the case in Mesosuchus" I'm not so sure. The "pneumatic" space (which I previously mentioned is continuous with the inner ear) is precisely where the lagena or cochlear duct would have been located. In fact the ventral tip of that space is at the BO/BSPH junction, which is typically where the lagena is directed. Perhaps in Mesosuchus the structure is there but is walled off laterally. It is indeed different and a little weird.

p. 21, lines 359-362: "Although a direct link between fossa volume and ecology has not been found for either extant birds or mammals (Ferreira-Cardoso et al., 2017; Walsh et al., 2013), in the absence of better proxies, it continues to be used to infer agility and modes of locomotion in extinct taxa (Bronzatti et al., 2017). Therefore, it will also be used herein." I'll quote from my previous review: "At a fundamental mechanistic level, the SCC and cerebellar flocculus are associated with sensing head and body movements and coordinating eye movements. Any connection to activity and posture must flow from the neurology, which is indirect at best in this case." I can also add now that any connection to ecology is also indirect at best. The authors cite Witmer et al. (2003 and 2008) as stating that scientists have related these systems to ecology and locomotion, but our emphasis as always been on their specific role in gaze stabilization mechanisms, which may relate to agility, locomotion, and ecology...but again only indirectly.

p. 21, lines 370-376: "The inner ear of Protoceratops shows an elongate common crus with relatively slender semicircular canals (Brown & Shclaikjer, 1940). The anterior and posterior canals are roughly the same length, while the lateral one is markedly shorter. The semicircular canals of Mesosuchus are as slender as those of Protoceratops, but the anterior canal is comparatively longer. The floccular lobe of Protoceratops is not developed, being absent from the endocast. Thus, if these structures reflect the mode of locomotion, Mesosuchus would have had a more erect posture and was probably more agile than Protoceratops." I'll again reiterate my criticism from my first review: "Comparison to Protoceratops—and certainly the inner ear of Protoceratops—are of doubtful utility given the vast difference in phylogenetic history." The authors' justification for maintaining this passage is not compelling. Moreover, the assumption that "these structures reflect the mode of locomotion," has not been substantiated. Again, they relate neurologically to coordinating eye movements.

---

## Round 0.3 · Minor Revisions

Dear Gabriela,

For the sake of reproducibility, the data used in your work should be made readily accessible via some more formal route than through a request to the authors. If a permanent URL link cannot be provided, access should at least be granted by formal application to the holding institution rather than by request to the researchers (which cannot ensure the long-term accessibility).

I am afraid I will not be able to consider your manuscript further until this requirement has been met.

Best regards,
Fabien Knoll

---

## Round 0.4 · accepted · Accept

Dear Gabriela,

I accept your manuscript for publication in PeerJ. I request, however, that you provide a more general contact on p. 4, one that ideally will be valid both in the near and distant future, when Johannes Müller leaves the museum. You can make this edit while in Production

Best regards,
Fabien Knoll

#